# Advances in retrieving $XCH_4$ and $XCO$ from Sentinel-5 Precursor: Improvements in the scientific TROPOMI/WFMD algorithm

Oliver Schneising, Michael Buchwitz, Jonas Hachmeister, Steffen Vanselow, Maximilian Reuter, Matthias Buschmann, Heinrich Bovensmann, and John P. Burrows

Institute of Environmental Physics (IUP), University of Bremen FB1, Bremen, Germany

**Correspondence:** O. Schneising (oliver.schneising@iup.physik.uni-bremen.de)

**Abstract.** The TROPOspheric Monitoring Instrument (TROPOMI) onboard the Sentinel-5 Precursor satellite enables the accurate determination of atmospheric methane ($CH_4$) and carbon monoxide ($CO$) abundances at high spatial resolution and global daily sampling. Due to its wide swath and sampling, the global distribution of both gases can be determined in unprecedented detail. The scientific retrieval algorithm Weighting Function Modified Differential Optical Absorption Spectroscopy (WFMD)

has proven valuable in simultaneously retrieving the atmospheric column-averaged dry air mole fractions $XCH_4$ and $XCO$ from TROPOMI's radiance measurements in the shortwave infrared (SWIR) spectral range.

     Here we present recent improvements of the algorithm which have been incorporated into the current version v1.8 of the TROPOMI/WFMD product. This includes processing adjustments such as increasing the polynomial degree to 3 in the fitting procedure to better account for possible spectral albedo variations within the fitting window and updating the digital elevation

model to minimise topography related biases. In the post-processing, the machine-learning-based quality filter has been refined using additional data when training the random forest classifier to further reduce scenes with residual cloudiness that are incorrectly classified as good. In particular, the cloud filtering over the Arctic ocean is considerably improved. Furthermore, the machine learning calibration, addressing systematic errors due to simplifications in the forward model or instrumental issues, has been optimised. By including an additional feature associated with the fitted polynomial when training the corresponding

random forest regressor, spectral albedo variations are better accounted for. To remove vertical stripes in the $XCH_4$ and $XCO$ data, an efficient orbitwise destriping filter based on combined wavelet-Fourier filtering has been implemented, while optimally preserving the original spatial trace gas features. The temporal coverage of the data records has been extended to the end of April 2022 covering a total length of 4.5 years since the start of the mission and will be further extended in the future.

     Validation with the ground-based Total Carbon Column Observing Network (TCCON) demonstrates that the implemented

improvements reduce the pseudo-noise component of the products resulting in an improved random error. The $XCH_4$ and $XCO$ products have similar spatial coverage from year to year including high latitudes and the oceans. The analysis of annual growth rates reveals an accelerated growth of atmospheric methane during the covered period, in line with observations at marine surface sites of the Global Monitoring Division of NOAA's Earth System Research Laboratory, which reported consecutive annual record increases over the past two years 2020 and 2021.

## 1 Introduction

Methane ($CH_4$) is the second most important greenhouse gas released by anthropogenic activity. Although it is less abundant, it exhibits a significantly larger mass-related global warming potential than carbon dioxide ($CO_2$), which is responsible for most of the human-induced radiative forcing since 1750 (Masson-Delmotte et al., 2021). Comprehensive knowledge of the source and sink processes of $CH_4$ is essential for a reliable prediction of future climate. Since $CH_4$ (with a lifetime of about 9 years) is considerably shorter-lived in the atmosphere than $CO_2$, reducing methane emissions offers the opportunity to take advantage of the corresponding short-term climate benefits to come within reach of the goal of limiting global warming to $1.5°C$. In this context, global detection and quantification of methane sources through satellite remote sensing can contribute to the emission mitigation efforts by identifying main emitters of anthropogenic origin, e.g. from the energy, waste or agricultural sectors, and thus suggesting effective emission reduction strategies.

Carbon monoxide (CO) is an air contaminant that is released into the atmosphere during combustion processes and as a result of the oxidation of hydrocarbons. This occurs both through natural processes and human activities. CO has adverse health effects as it impairs the oxygen-carrying capacity of the blood by directly binding to haemoglobin (Rose et al., 2015). Due to its lifetime of approximately 1-2 months, it is well suited as tracer of the long-range transport of pollutants. In the presence of sufficient $NO_x$, CO participates in the net production of tropospheric ozone (Fowler et al., 2008), which is a greenhouse gas and known to be harmful to health. In addition, CO is the main sink for the hydroxyl radical (OH) reducing the potential of atmospheric self-cleansing because the chemically depleted OH is no longer available to remove other constituents of the atmosphere, including greenhouse gases such as methane. Consequently, CO is considered an indirect contributor to climate change.

Continuous global satellite observations of both gases enable a better insight into atmospheric transport, tropospheric chemistry, and the climate system. Measurements of the upwelling radiances in the shortwave infrared (SWIR) spectral region have been exploited for the retrieval of the abundances and distributions of trace gases, such as $CO_2$, $CH_4$, and CO, because they are sensitive to all atmospheric layers. The Measurement of Pollution in the Troposphere (MOPITT) instrument (Drummond et al., 2010) onboard NASA's Terra satellite successfully retrieves CO from both the thermal and shortwave infrared and was intended to measure $CH_4$. The SCanning Imaging Absorption spectroMeter for Atmospheric CHartographY (SCIAMACHY) (Burrows et al., 1995; Bovensmann et al., 1999) onboard ESA's ENVISAT began simultaneous measurements of $CO_2$, $CH_4$, and CO from space (Buchwitz et al., 2005; Frankenberg et al., 2006). The Thermal And Near infrared Sensor for carbon Observations Fourier Transform Spectrometer (TANSO-FTS) onboard GOSAT (Kuze et al., 2016) observes $CO_2$ and $CH_4$ absorption lines, its successor TANSO-FTS-2 onboard GOSAT-2 (Suto et al., 2021) additionally measures CO due to an extended spectral range. The previously existing application areas of satellite data were further expanded with the TROPOspheric Monitoring Instrument (TROPOMI), which is considered groundbreaking for determining atmospheric composition, including $CH_4$ and CO, from space with respect to combined spatio-temporal coverage and data quality.

TROPOMI is the only payload instrument of the ESA Sentinel-5 Precursor mission launched in October 2017 (Veefkind et al., 2012). It is a pushbroom imaging spectrometer measuring radiances in eight spectral bands from the ultraviolet (UV) to

the shortwave infrared (SWIR) allowing the retrieval of various atmospheric constituents. TROPOMI has the unique ability to combine observations at high spatial resolution with global coverage on a daily basis due to its large $2600\,\mathrm{km}$ swath consisting of individual measurements with a footprint size of $5.5 \times 7\,\mathrm{km}^2$ at nadir ($7 \times 7\,\mathrm{km}^2$ before 6 August 2019) in the SWIR bands (7 and 8) relevant for the methane and carbon monoxide retrieval. As a consequence, both gases can be detected worldwide in unprecedented detail. If desired, the globally available TROPOMI data may be complemented in a symbiotic way by targeted remote sensing data from aircraft or satellites with higher spatial resolution but limited coverage, e.g. from GHGSat (Cusworth et al., 2021), to detect and identify significant emitters.

In addition to the operational TROPOMI products for $CH_4$ (Hu et al., 2016; Hasekamp et al., 2022) and CO (Landgraf et al., 2016, 2022), there is also a scientific methane product based on the same algorithm as the operational product but with optimised settings (Lorente et al., 2021, 2022), as well as the scientific WFMD product providing both gases simultaneously retrieved from the same fitting window (Schneising et al., 2019; Schneising, 2022). The scientific products have turned out to be valuable in verifying or improving the operational products and in assessing the robustness of results by analysing the extent to which specific results are sensitive to the details of the algorithm setup.

Here we introduce the recent changes incorporated in the latest product version of TROPOMI/WFMD. After a short review of the basics of the retrieval algorithm including machine learning quality filter and bias correction, the following sections describe in detail the improvements implemented in the current version and demonstrate their benefits. Finally, an overview of the new improved products is provided including a discussion of atmospheric annual growth rates and validation results compared to previous product versions.

## 2 TROPOMI/WFMD retrieval algorithm

The scientific retrieval algorithm Weighting Function Modified DOAS (WFMD) for simultaneously retrieving the atmospheric column-averaged dry air mole fractions of methane ($XCH_4$) and carbon monoxide ($XCO$) from radiance measurements in the SWIR of the TROPOMI instrument onboard the Sentinel-5 Precursor satellite (S5P) is described in detail in Schneising et al. (2019). It is a least-squares procedure using scaling (or shifting) of previously selected atmospheric vertical profiles (Buchwitz et al., 2007; Schneising et al., 2011). The linearised radiative transfer model (based on the tabulation of radiances and associated derivatives obtained from the radiative transfer model SCIATRAN (Rozanov et al., 2002, 2014) in pseudo-spherical atmosphere mode) is fitted together with a low degree polynomial to the logarithm of the measured sun-normalised radiance. The look-up table enables a fast retrieval of the vertical columns of the targeted species and contains various reference spectra characterising a variety of typical atmospheric conditions covering combinations of different solar zenith angles, altitudes, albedos, water vapour contents, and temperatures.

Given the number $m \in \mathbb{N}$ of spectral points to be fitted and the number $n \in \mathbb{N}$ of state vector elements (with $m \gg n$), the best fit of the linearised model to the observed radiance is obtained by minimising

$$f(\boldsymbol{x}) = \left\| \mathbf{W}^{\frac{1}{2}} \left( \boldsymbol{y} - \mathbf{A}\boldsymbol{x} \right) \right\|_2^2 = \left( \boldsymbol{y} - \mathbf{A}\boldsymbol{x} \right)^{\mathrm{T}} \mathbf{W} \left( \boldsymbol{y} - \mathbf{A}\boldsymbol{x} \right) \tag{1}$$

with respect to the difference $x \in \mathbb{R}^n$ of the state vector and the multidimensional linearisation point. Here $y \in \mathbb{R}^m$ is the corresponding difference vector of measurement and linearised model, $\mathbf{A} \in \mathbb{R}^{m \times n}$ is the Jacobian (matrix of weighting functions, i.e. derivatives at the linearisation point, and polynomial basis functions as columns), and $\mathbf{W} = \mathbf{C_y}^{-1}$ the matrix of weights defined as the inverse of the measurement noise covariance matrix $\mathbf{C_y} = \mathrm{diag}(\sigma_1^2, \ldots, \sigma_m^2) \in \mathbb{R}^{m \times m}$ associated with uncorrelated measurement noise $\sigma_i$ at the different spectral points. This minimisation yields the solution $\hat{x} = \mathbf{C_x}\mathbf{A}^{\mathrm{T}}\mathbf{W}y$ of the inverse problem, with $\mathbf{C_x} = \left(\mathbf{A}^{\mathrm{T}}\mathbf{W}\mathbf{A}\right)^{-1}$ being the covariance matrix of solution $\hat{x}$.

In post-processing, the retrieved vertical columns are converted to column-averaged dry air mole fractions (denoted $XCH_4$ and XCO) by dividing them by the dry air columns obtained from European Centre for Medium-Range Weather Forecasts (ECMWF) data. To adopt the high spatial resolution of the TROPOMI data, the ECMWF dry air columns are adjusted for the actual elevation of the individual satellite scenes by accounting for the mismatch to the mean altitude of the coarser model grid.

As the look-up table is confined to specific atmospheric conditions such as cloud-free scenes, an efficient machine learning quality filtering procedure based on a random forest classifier (Schneising et al., 2019) is trained using quasi-simultaneous cloud information from the Visible Infrared Imaging Radiometer Suite (VIIRS) onboard Suomi NPP (Hutchison and Cracknell, 2005) flying in a constellation 3.5 minutes ahead of the Sentinel-5 Precursor. Due to the strict quality requirements for retrieving atmospheric methane, a shallow learning random forest regressor (shallow refers here to the number of leaves in each decision tree being small compared to the cardinality of the training dataset) is trained to reduce the remaining systematic coarse-scaled methane dependencies on other parameters such as albedo after quality filtering. This is done by assuming the Simple cLImatological Model for atmospheric $CH_4$ (SLIMCH4) v2021 (Noël et al., 2022) without interpolation as a sound low-resolution ($6° \times 4°$) approximation of truth. This correction only compensates for gross statistical discrepancies regarding the considered features (mainly albedo-related), as the tree growing is limited and the training is restricted to a subperiod of time and a few small regions, which are selected to cover the whole range of albedo values and all possible viewing geometries. Therefore, the training data set is considered representative of the entire globe with respect to albedo-related biases, ensuring a reliable global correction. The additional pruning of the decision trees is performed to keep the correction as simple, fast, and shallow as possible to be certain that potential specifics of the climatology are not over-learned. A similar calibration of XCO is not necessary to achieve the mission requirements since a potential albedo-induced bias of the same percentage magnitude as for $XCH_4$ would not be significant due to the considerably higher variability of XCO.

## 3   Algorithm improvements

Since version v1.2 (Schneising et al., 2019), several changes have been implemented in the WFMD algorithm, which are described in this section and summarised in Table 1. There are rather straightforward adjustments, such as increasing the resolution of the used gridding of the underlying topography database from $0.05°$ to $0.025°$ and using the ECMWF Reanalysis v5 (ERA5, with resolution $0.25° \times 0.25° \times 1h$) (Hersbach et al., 2020) instead of the ECMWF analysis ($0.75° \times 0.75° \times 6h$) in the post-processing since v1.5. The theoretical improvement due to the better temporal and spatial resolution of the meteorological data is difficult to demonstrate and typically small, because noticeable impact is only expected when conditions

**Table 1.** Overview of the different TROPOMI/WFMD versions and the respective differences in the setup. For details see the main text.

| | v1.2 | v1.5 | v1.6 | v1.7 | v1.8 |
|---|---|---|---|---|---|
| **Meteorological data** | ECMWF analysis $0.75° \times 0.75° \times 6h$ | ECMWF Reanalysis v5 (ERA5) $0.25° \times 0.25° \times 1h$ | | | |
| **Polynomial degree** (Section 3.1) | 2 | | 3 | | |
| **Digital Elevation Model** (Section 3.2) | GMTED2010 $0.05° \times 0.05°$ | GMTED2010 $0.025° \times 0.025°$ | | GMTED2010 + ICESat-2 Greenland $0.025° \times 0.025°$ | Copernicus GLO-90 $0.025° \times 0.025°$ |
| **Quality Filter** (Section 3.3.1) | Random Forest Classifier Training with 16 days and 25 features | Random Forest Classifier Training with 30 days and 26 features (as before + surface roughness) + 3-step filter | | as previous version + ocean refinement | |
| **Methane Calibration** (Section 3.3.2) | Random Forest Regressor 5 features in training: Albedo, SZA, $r_{cld}$, $L_{sH_2O}$, $i_{acr}$ 500 leaf nodes Training limited to SZA$< 75°$ | Random Forest Regressor 6 features in training: as before + $p_0$ 5000 leaf nodes Training limited to SZA$< 75°$ | | as previous version, but 10000 leaf nodes Training limited to SZA$< 80°$ | |
| **Destriping** (Section 3.4) | No | | | Yes | |
| **Reference** | Schneising et al. (2019) | Schneising (2022) | - | Hachmeister et al. (2022) | presented here |

change significantly on a small scale and/or in the short term. The more comprehensive and distinct improvements are described in the following subsections.

## 3.1 Polynomial fit parameters

In the original settings a quadratic function (polynomial of degree 2) was used in the minimisation procedure of (1) to take broadband effects into account when fitting the linearised radiative transfer model to the observed radiance. Although surface albedo has been addressed as a potential source of biases from the beginning in the TROPOMI/WFMD algorithm, residual suspect signals were retrieved in rare cases even in v1.5 over areas exhibiting special surface characteristics. This finding was attributed to the variability of spectral albedo within the used fitting window around $2.3\,\mu m$ for specific soil types depending on their exact chemical composition. Potential interferences in this spectral range could be, for example, spectral features due to OH-metal bend and OH stretch combinations (Tayebi et al., 2017) or features caused by combination tones of certain salts and gypsum (Moreira et al., 2014). Therefore, the polynomial degree in the fitting procedure was increased to 3 since v1.6 to better account for possible spectral albedo variations within the fitting window.

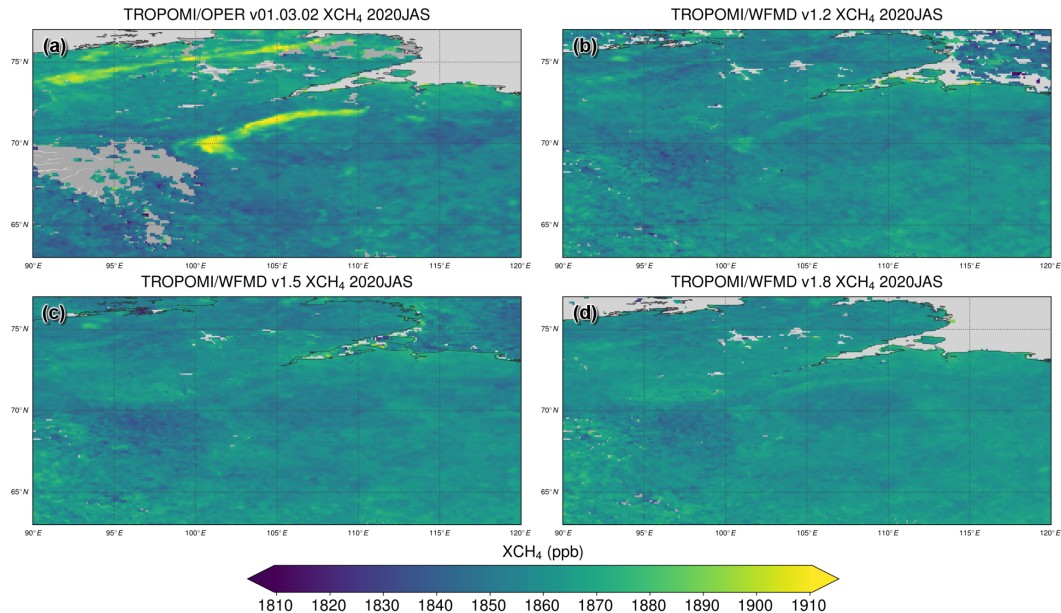

**Figure 1.** Comparison of the $XCH_4$ distribution in July-September 2020 over the Northern Siberian Taymyr region for **(a)** the operational product and **(b-d)** different TROPOMI/WFMD versions (v1.2 and v1.5 use a quadratic polynomial, while v1.8 uses a cubic polynomial). The strong artificial enhancement in the operational product, that is largely reduced when using a cubic polynomial (Lorente et al., 2022), is hardly visible at all in the TROPOMI/WFMD products even if using a quadratic polynomial.

Concurrent with our analysis, Lorente et al. (2022) also investigated the impact of the polynomial degree on their scientific TROPOMI methane product, which is based on the RemoTeC algorithm just like the operational product, finding significant improvements for several regions when increasing the polynomial degree from 2 to 3. However, the impact of this modification on their product is quite different, both qualitatively and quantitatively, than for TROPOMI/WFMD due to the alternative
algorithm setup, e.g. with differing fitting windows and bias correction schemes. An example of this is the artefact in the Northern Siberian Taymyr region that occurs in the operational product (Barré et al., 2021), which was also misinterpreted as genuine large methane emissions from carbonate rock formations (Froitzheim et al., 2021). While Lorente et al. (2022) demonstrate that the original strong enhancement in the operational and RemoTeC scientific product is not reproduced when using a cubic polynomial, a corresponding distinct enhancement in this magnitude is not observed in the TROPOMI/WFMD
products, even in the previous setup with a quadratic polynomial (see Figure 1).

Overall, the changes due to the adjustment of the polynomial degree seem to be less significant for TROPOMI/WFMD than for RemoTeC. The lower susceptibility of TROPOMI/WFMD to some of the observed spectral albedo bias features is primarily attributed to the narrower spectral fitting windows in comparison to the RemoTeC retrievals, which cover a wider spectral range. If the spectral range is sufficiently small, it is easier to approximate albedo-induced structures in the spectral baseline by lower
degree polynomials. To retrieve $CH_4$ and CO simultaneously as accurately as possible, the TROPOMI/WFMD spectral fitting

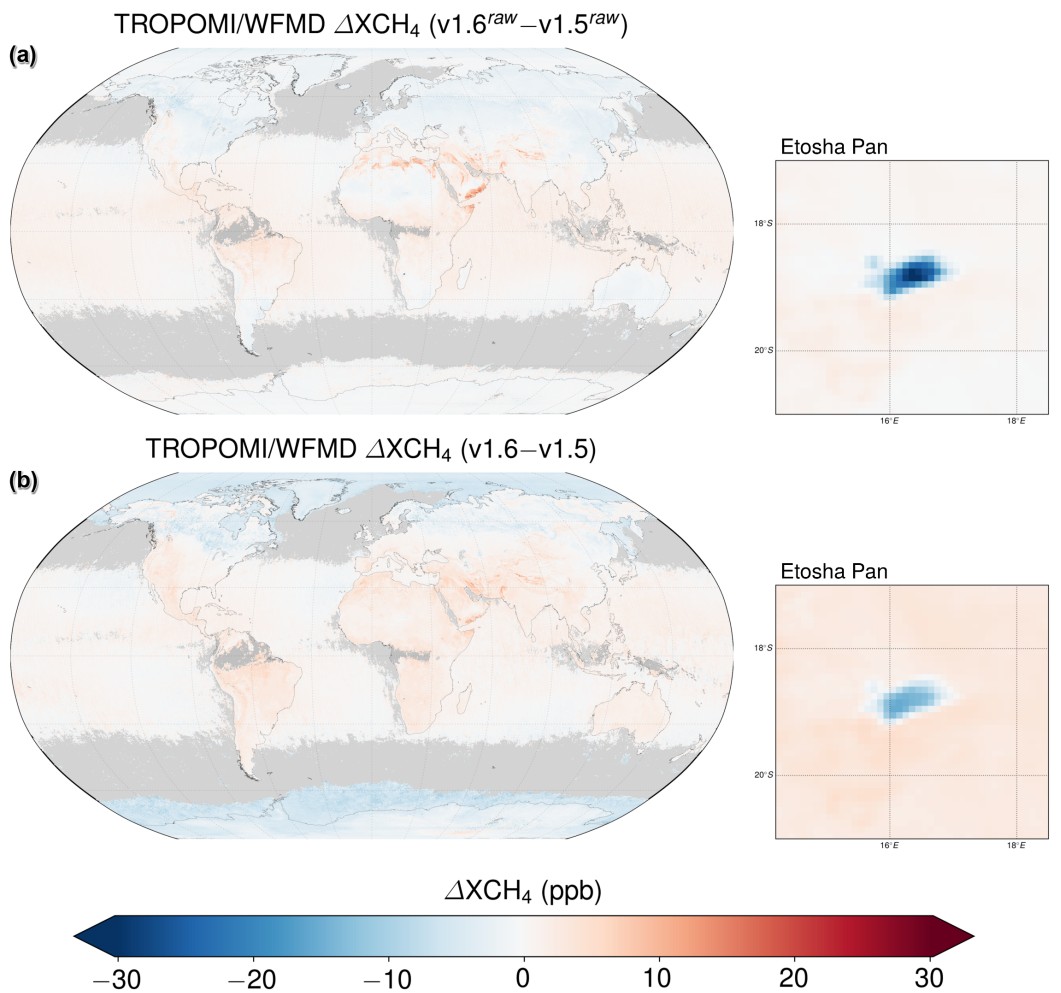

**Figure 2.** Global differences in $XCH_4$ (2018-2020) when changing the polynomial degree from 2 to 3 **(a)** before and **(b)** after application of the machine learning calibration. The largest differences are observed over the Etosha pan. The differences in Northern Africa and on the Arabian peninsula are largely resolved via the calibration when using a quadratic polynomial as these features are mitigated in the lower panel.

windows were optimised based on an error analysis of simulated measurements (also including spectral albedo scenarios of typical surface types) resulting in the windows 2311–2315.5 nm and 2320–2338 nm (Schneising et al., 2019). For instance, it was identified that it is beneficial with regard to systematic errors to exclude the strong methane absorption lines between the two fitting windows, although the associated loss of spectral information may lead to a slightly increased random error.

The global differences resulting from the adjustment of the polynomial degree are depicted in Figure 2 before and after application of the machine learning calibration. Both versions shown (v1.5 and v1.6) only differ by the polynomial degree used in the fitting procedure according to Table 1. The quality filter and calibration is then performed individually for both versions

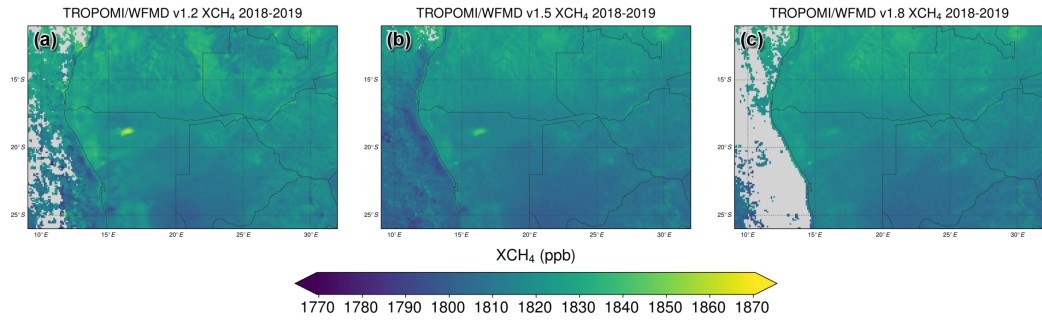

**Figure 3.** Comparison of the $XCH_4$ distribution over Southern Africa for **(a)** v1.2, **(b)** v1.5 (both using a quadratic polynomial), and **(c)** v1.8 using a polynomial of degree 3 instead in the fitting procedure to better account for potential variability of spectral albedo within the fitting window. As a consequence, the enhancement over the Etosha pan in Namibia is significantly reduced and considered more realistic (see main text for details). The differences in coverage over the ocean are due to the stricter quality filter in v1.8 (see Section 3.3). There is already a noticeable improvement from v1.2 to v1.5 due to the improved calibration taking spectral albedo variability at least partially into account by adding the polynomial parameter $p_0$ as feature.

but with the same settings in each case (e.g. the list of features available for selection for the random forest regressor). Figure 2a demonstrates the direct impact of using a cubic polynomial instead of a quadratic one. The largest impact is observed over the Etosha pan in Namibia. Further notable differences appear in Northern Africa and on the Arabian peninsula. Figure 2b allows to identify which of the occuring differences are alternatively resolved by the subsequent calibration when using a quadratic polynomial: This is the case when a localised pattern appearing in the upper panel is mitigated in the lower panel. Thus, the changes due to the cubic polynomial in Northern Africa and on the Arabian peninsula were achieved by the machine learning calibration in earlier versions and there is no or only a small effective impact in the final product for these regions. Nevertheless, the overall calibration has the potential to work somewhat better in the cubic case, since the aforementioned specific regional spectral albedo features do not need to be explicitly addressed any more and other statistical coherences may then be better identified. In contrast, however, the differences over the Etosha pan cannot be entirely achieved through calibration justifying the implementation of the increased polynomial degree. This is even more true for XCO, which shows similar regional changes through the increase in polynomial degree, but is not calibrated due to higher variability and weaker requirements compared to $XCH_4$.

As the Etosha pan is also the region where the most prominent potential bias in previous versions was observed, we analyse the improvement due to the increase of the poynomial degree for this special case in more detail. The Etosha pan is a large endorheic salt pan in Namibia. It exhibits intermittent shallow inundation and is therefore considered a wetland and thus a potential methane source. After the infiltrated water has evaporated again, a salt crust remains on the ground. According to Beugler-Bell and Buch (1997), the soils of the Etosha pan are calci sodic Solonchaks to sali calcic Solonetzs derived from Andoni sandstone or siltstone, which may result in an overestimation of the detected methane enhancement, if the fitted quadratic polynomial is not able to approximate the spectral albedo of this specific soil type sufficiently well. As a result of the

increase of polynomial degree to 3, the enhancement over the Etosha pan is significantly reduced (see Figure 3). As the other spatial features are preserved, the modification of the polynomial degree is considered a general improvement, with positive effects in the presence of specific soil types and otherwise no significant changes compared to previous product versions. The comparison with v1.2 in Figure 3 also confirms the finding of Figure 2 that a part of the enhancement can be resolved by the calibration, which has been improved compared to v1.2 by adding the polynomial parameter $p_0$ as feature (see Table 1) to take spectral albedo variability at least partially into account.

The interpretation that the distribution over the pan is more realistic when using a cubic polynomial is based on the fact that the shape of the retrieved enhancement for single overpasses is more variable. While the enhancement reflects the extent of the pan virtually always in the case of a quadric polynomial, it changes with meteorological conditions in the cubic case. For instance, the link between inundation and methane enhancement appears more evident in the latest product version. Instead of a fixed shape, the enhancement is now more pronounced when parts of the pan are shallowly flooded, as opposed to days with utter drainage. This is demonstrated in Figure 4 by comparison with a VIIRS false colour band combination (Red = M3 $(478 - 498\,\mathrm{nm})$, Green = I3 $(1580 - 1640\,\mathrm{nm})$, Blue = M11 $(2225 - 2275\,\mathrm{nm})$) distinguishing different water states and thus enabling flood mapping. In this band combination, vegetation appears in green, bare soil or deserts in bright cyan, and liquid water on the ground or sediments in water will appear dark. The figure compares days from the dry seasons 2019 and 2020, which differ significantly in terms of flooding. Due to abundant rainfall in the second half of the wet season (December 2019 - March 2020) the eastern part of the pan stayed inundated until end of July in the year 2020, well into the dry season. In contrast, the pan was entirely drained during the dry season 2019. Consistent with this, there is a methane enhancement over the eastern pan in 2020 that is not observed in 2019 in the TROPOMI/WFMD v1.8 product. For comparison, Figure 4 also shows the operational product, which, however, has virtually no data over the Etosha pan on the analysed days. Thereby, like in all other comparisons with the operational product, only data with quality assurance level larger than the recommended value (qa_value $> 0.5$) is shown.

The improvement due to the increased polynomial degree is also reflected in an improved fit quality. Figure 5 shows fits for two example scenes inside and outside of the Etosha pan and maps of the region demonstrating the fit quality for quadratic and cubic polynomials from the day in the dry season 2019 depicted in Figure 4, where the pan was utterly drained. As can be seen, the fit quality improves significantly over the pan when increasing the polynomial degree from 2 to 3 and the root mean square of the fit residual $\epsilon_{RMS}$ becomes comparable to the surroundings of the pan, while there is no significant change outside of the pan. In addition to the map of the RMS of the fit residuals, the associated XCO map is also much more homogeneous and the outlines of the pan are no longer visible when using a cubic polynomial. The sample $XCH_4$ values of the two analysed scenes converge to a common value inside and outside of the pan in both the raw and calibrated data. The respective bias in the quadratic case of the calibrated data of about $24\,\mathrm{ppb}$ virtually vanishes completely. Based on the analyses in this section and due to the identified homogeneity in maps of $XCH_4$, XCO, and fit quality, the retrieved values over the identified main problem area Etosha pan (and everywhere else in the world) are considered realistic when using a cubic polynomial.

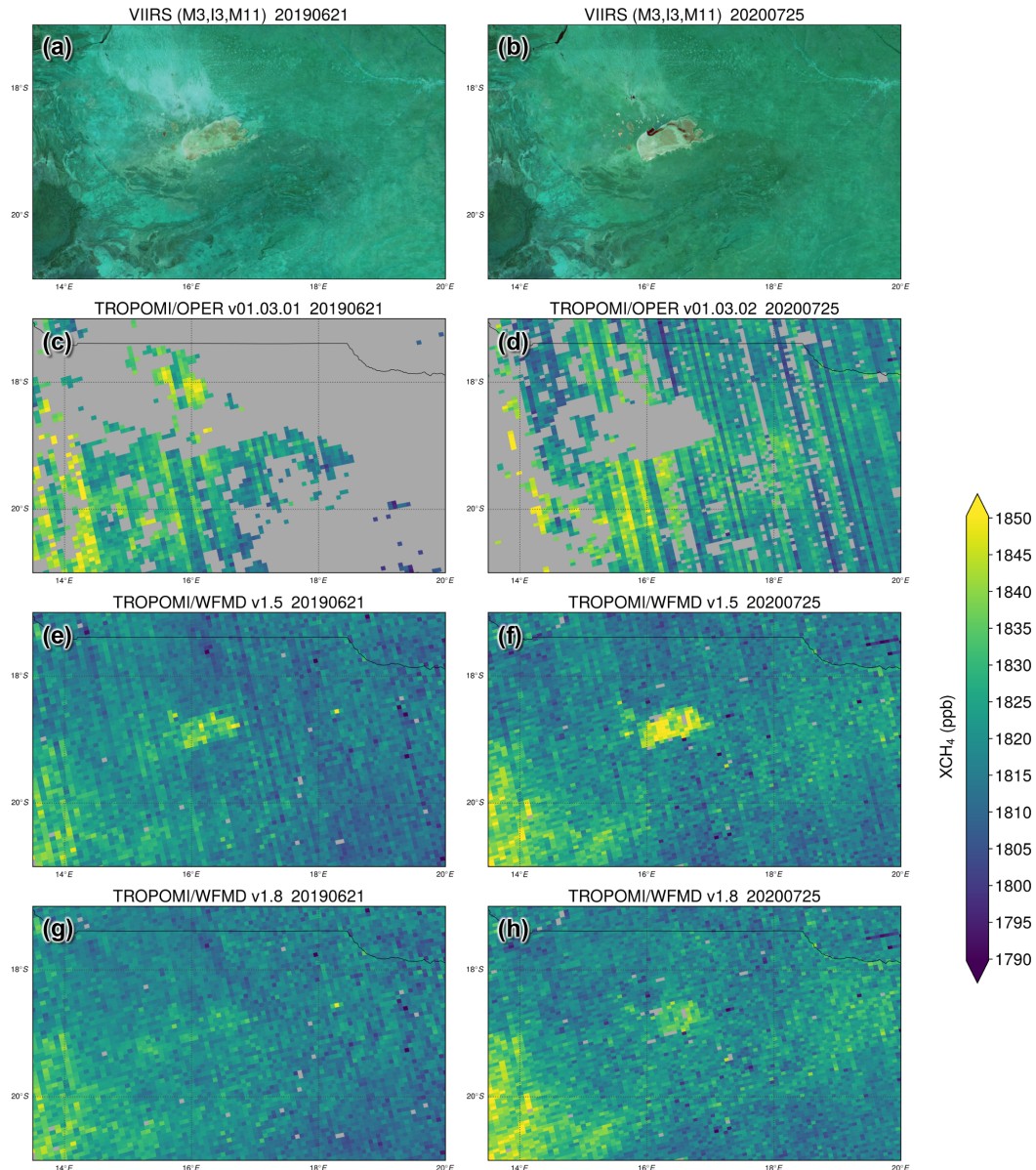

**Figure 4.** Comparison of a **(a-b)** VIIRS false colour band combination distinguishing different water states (taken from the NASA Worldview application) with the $XCH_4$ distribution over the Etosha pan for **(c-d)** the operational product, **(e-f)** TROPOMI/WFMD v1.5 and **(g-h)** TROPOMI/WFMD v1.8 on two different days **(a,c,e,g)** without and **(b,d,f,h)** with shallow inundation of parts of the pan, which appear dark in the VIIRS false colour image. In the latter case there is a perceivable enhancement in the swamped portion for v1.8, which does not arise on the drained day. Please note that for v1.8 an additional destriping algorithm has been applied (see Section 3.4).

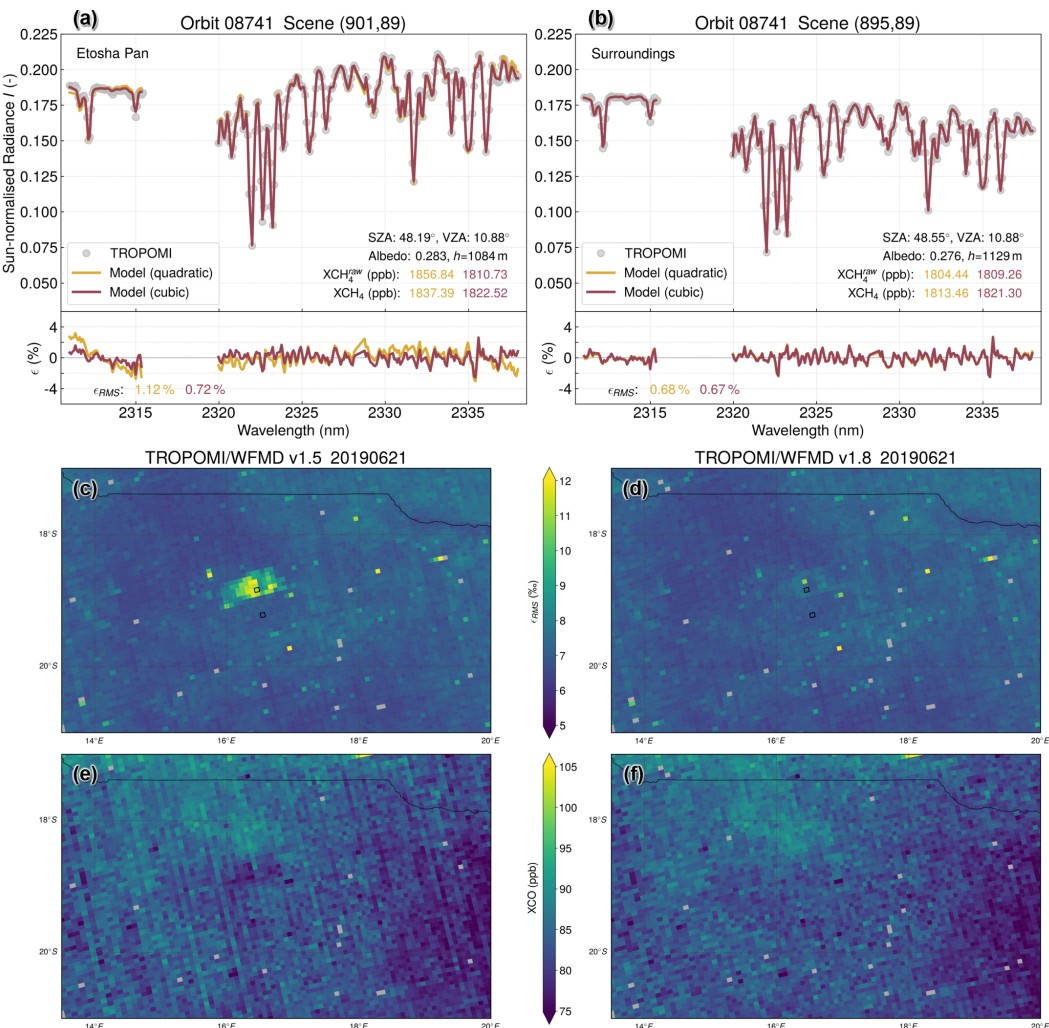

**Figure 5.** Example fits for two scenes **(a)** inside and **(b)** outside of the Etosha pan. The spectral TROPOMI measurements are shown as gray dots, the respective models (and fit residuals) are shown in yellow and red for using a quadratic or cubic polynomial in the fitting procedure. Also shown are maps for a single satellite overpass of **(c-d)** the root mean square of the fit residuals (with the two example scenes being highlighted by a black border) and **(e-f)** XCO. The maps **(d,f)** associated with a cubic polynomial (v1.8) are much more homogeneous and the extent of the pan is no longer visible. Please note that for v1.8 XCO an additional destriping algorithm has been applied (see Section 3.4).

## 3.2 Digital elevation model

The previously used Digital Elevation Model (DEM) Global Multi-Resolution Terrain Elevation Data 2010 (GMTED2010) (United States Geological Survey, 2018) was replaced in WFMD v1.8 as significant local inaccuracies were identified over Greenland compared to ICESat-2 data (Hachmeister et al., 2022). The associated analysis confirmed the expected linear cor-
relation between $\Delta h$ and $\Delta \mathrm{XCH_4}$. Thereby, an error of $1\%$ in the assumed surface pressure corresponds approximately to an error of $1\%$ in retrieved $\mathrm{XCH_4}$. As a universally consistent current DEM is preferable in the case of a global satellite dataset and inaccuracies of GMTED2010 also affect other regions of the globe, the Copernicus GLO-90 DEM (European Space Agency, 2021) was utilised in the updated product (instead of the ICESat-2 data used in Hachmeister et al. (2022) for the Greenland case study) to minimise topography related biases in the TROPOMI/WFMD $\mathrm{XCH_4}$ and XCO data globally.

The freely available Copernicus GLO-90 DEM is based on radar satellite data collected between 2011-2015 during the TanDEM-X Mission and represents the full global landmass including buildings, infrastructure and vegetation at a resolution of $90\,\mathrm{m}$. A global validation with ICESat-1 data provides an average vertical accuracy of $2.17\,\mathrm{m}$ at $90\%$ confidence level (Airbus, 2020). In this global assessment, Greenland and Antarctica were excluded from the statistics due to the additional uncertainties associated with snow and ice cover. Broken down by ecoregion, the accuracy for most surface types is better than $2\,\mathrm{m}$, while tundra, boreal forests, and woodlands have the lowest accuracy levels compared to other regions. This lower accuracy is expected because the penetration depths of radar and lidar differ for regions covered with conifers. A similar penetration depth issue exists for dry firm snow hampering a direct comparison with ICESat-1 data for regions with permanent snow or ice cover leading to lower accuracy levels in these regions. Consequently, the obtained vertical accuracies for Greenland and Antarctica are about three times higher than for the global assessment ($7.26\,\mathrm{m}$ and $6.38\,\mathrm{m}$ at $90\%$ confidence level).

To get an idea of the impact of the DEM change, Figure 6 shows the average difference in $\mathrm{XCH_4}$ when substituting GMTED2010 with GLO-90. As can be seen, the differences are rather small for most regions of the world, but can become large at individual locations, e.g. in Artic regions (parts of Greenland and Spitsbergen) or Antarctica, where biases associated with singular DEM inaccuracies of a few $100\,\mathrm{m}$ could exceed $100\,\mathrm{ppb}$ before the DEM change. There is also an increase in the inhomogeneity of the difference with a discontinuous appearance at high latitudes above $60°\mathrm{N}$, in particular in Eurasia, which can be attributed to the GMTED2010 data. This is a known limitation of GMTED2010, as the DEM is composed of various datasets and the main dataset used in GMTED2010, the Digital Terrain Elevation Data v2 (DTED 2), is only available for latitudes between $60°\mathrm{N}$ and $56°\mathrm{S}$ (Danielson and Gesch, 2011). The topography information at higher latitudes is based on older data with lower spatial resolution. That the high latitude difference pattern is an issue of GMTED2010 and not of GLO-90 can also be seen in Figure 1 showing the methane distribution for several TROPOMI/WFMD products over Northern Siberia demonstrating that v1.8, which is based on GLO-90, exhibits the most homogeneous methane distribution in this region. The inference that the updated product features a more consistent and realistic methane distribution is also evident over Greenland, where issues of GMTED2010 have already been identified by Hachmeister et al. (2022) using ICESat-2 data. The improvement through the use of GLO-90 is demonstrated in Figure 7. With respect to the TROPOMI data, a similar improvement is

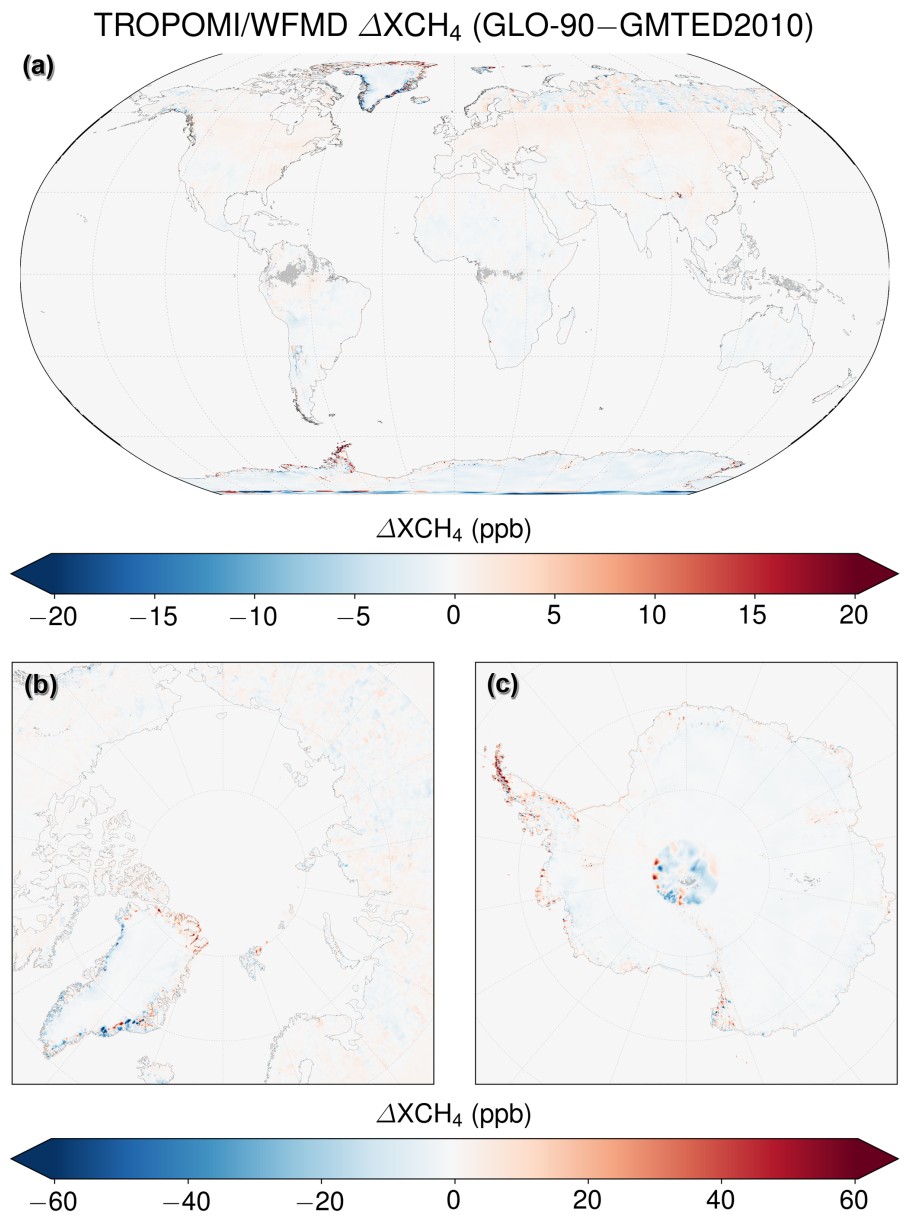

**Figure 6.** Global differences **(a)** in $XCH_4$ when substituting the GMTED2010 digital elevation model with GLO-90. The largest differences are in the **(b)** Arctic, in particular Greenland, and **(c)** Antarctica.

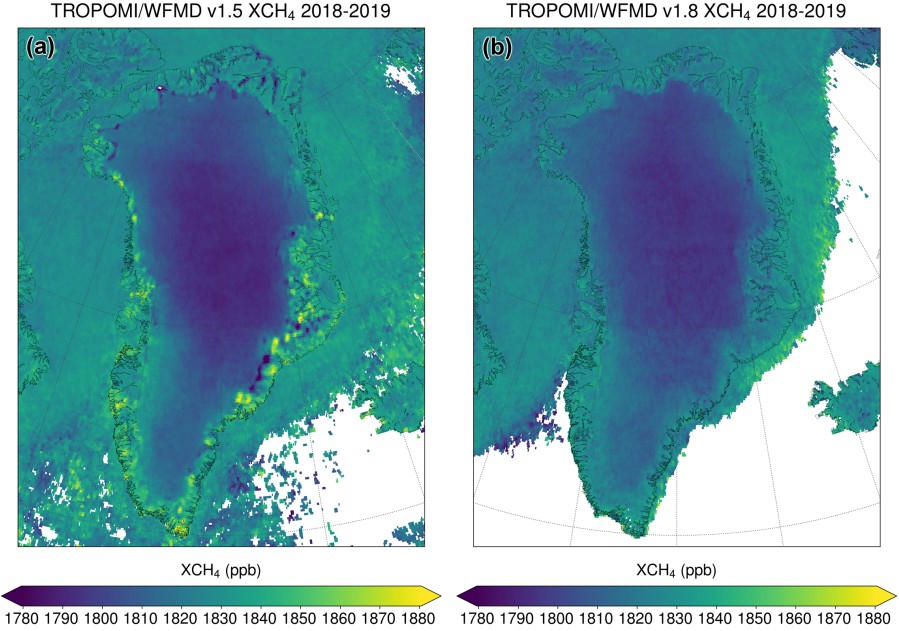

**Figure 7.** Comparison of the XCH$_4$ distribution over Greenland for **(a)** v1.5 based on GMTED2010 and **(b)** v1.8 using updated topography. The suspicious features, which are correlated with the topography-related differences shown in Figure 6, disappear when using the GLO-90 digital elevation model. The differences in coverage over the ocean are due to the stricter quality filter in v1.8 (see Section 3.3).

obtained over Greenland when using GLO-90 or ICESat-2 instead of GMTED2010. The added value of GLO-90 is its global
consistency, which additionally permits to resolve further potential DEM inaccuracies and related retrieval biases elsewhere.

Regarding biases induced by DEM issues, there are similar percentage improvements for the latest XCO product, but the previous inaccuracies were less significant due to the higher variability of XCO and the lower requirements on the product compared to XCH$_4$.

## 3.3 Quality filter and posterior correction

The TROPOMI/WFMD algorithm uses machine learning in the post-processing to predict low-quality scenes and to calibrate the methane retrievals by learning statistical relationships. In addition to the dedicated changes to the settings in the machine learning framework described in this section and summarised in Table 1, further effective improvements may arise if the actual underlying systematic dependencies are better teased out due to the other refinements, such as improved topography or updated polynomial degree.

### 3.3.1 Machine learning quality filter

For the fastest possible Level 1 to Level 2 processing, the look-up table of radiances and their derivatives with respect to the state vector elements should be kept appropriately small. As a consequence, some assumptions and simplifications concerning

atmospheric conditions have to be made resulting in the need to exclude measurements that are not adequately represented by the forward model. This primarily applies to cloudy scenes, which have to be rigorously filtered out, as the look-up table is limited to cloud-free conditions. To this end, a random forest classifier based on machine learning was implemented in the post-processing exploiting concurrent cloud information from the VIIRS instrument (Schneising et al., 2019). After the initial training, the algorithm is independent of the availability or consistency of the VIIRS data, as the classification is then entirely driven by intrinsic parameters available from or used in the preceding processing excluding the primary retrieval parameters XCH$_4$ and XCO.

To optimise coverage and quality of the products, the dataset used in the training of the random forest quality screening was extended to 30 randomly chosen days from end of April 2018 until end of 2019 since v1.5 (compared to the original 16 days). A total of 5 million measurements are selected for each day including all land data, all inland water data, and all ocean data passing the quality filter; the remaining amount is randomly sampled with bad quality ocean scenes. Furthermore, surface roughness (determined from the respective DEM by using the standard deviation of the high resolution data within the considered gridboxes) was included as an additional feature of the classifier to potentially allow better identification of scenes that may have quality deficits due to the simplifications of the forward model. These changes result in fewer residual outliers remaining after quality filtering.

Residual outliers due to issues not explicitly considered in the training of the quality filter are addressed by a heuristical 3-step quality filter that was additionally implemented (since v1.5) after application of the random forest classifier: 1) Retrievals whose shift and squeeze parameters exceed the $3\sigma$-range centred around the respective daily mean values are discarded. 2) An empirical filter with respect to the root mean square of the fit residual $\epsilon_{RMS}$ is applied depending on the sun-normalised radiance in a spectral range with negligible absorption $I_0$ at the edge of the fitting window. The quality is considered bad if $\epsilon_{RMS} > 0.027$ or $\epsilon_{RMS} > a \cdot (I_0 + b)^{-1} + c$ with parameters $a = 0.0019$, $b = 0.075$, $c = 0.007$, which are adjusted for scenes over ocean and inland water (i.e. with land fraction zero) to $a = 0.00063$, $b = 0.015$, $c = 0.009$. These parameters were determined empirically to distinguish between typical values of $\epsilon_{RMS}(I_0)$ and outliers. This filter step serves to exclude exceptional scenes with reduced fit quality relative to scenes with similar radiance for whatever reason such as specific scenes with intense aerosol exposure. 3) Unsupervised outlier detection is applied on daily data based on the comparison of the local density of a sample with the local densities of its neighbours. This is done by assigning a degree of being an outlier to each object in multidimensional space, called the Local Outlier Factor (LOF) (Breunig et al., 2000). It is local in the sense that the LOF depends on how isolated an object is from its surroundings, with LOF being close to 1 for objects inside a cluster. Here we use the 3-dimensional (latitude,longitude,XCH$_4$)-space and the standard Euclidean metric to measure the spatial distance. Thereby, any upward outliers regarding XCH$_4$ are not filtered out to avoid incorrectly excluding genuine emission point sources. The full 3-step procedure removes a further $3.5\%$ of the scenes that pass the random forest classifier quality screening algorithm, with about $3\%$ of the data filtered out by the shift/squeeze/$\epsilon_{RMS}$-Filter and about $0.5\%$ by the LOF outlier detection. Since this post-random-forest filter is only about remaining outliers it is impractical to identify and explicitly consider all conceivable issues in the training of the random forest classifier and this heuristic approach seems to be a satisfactory solution.

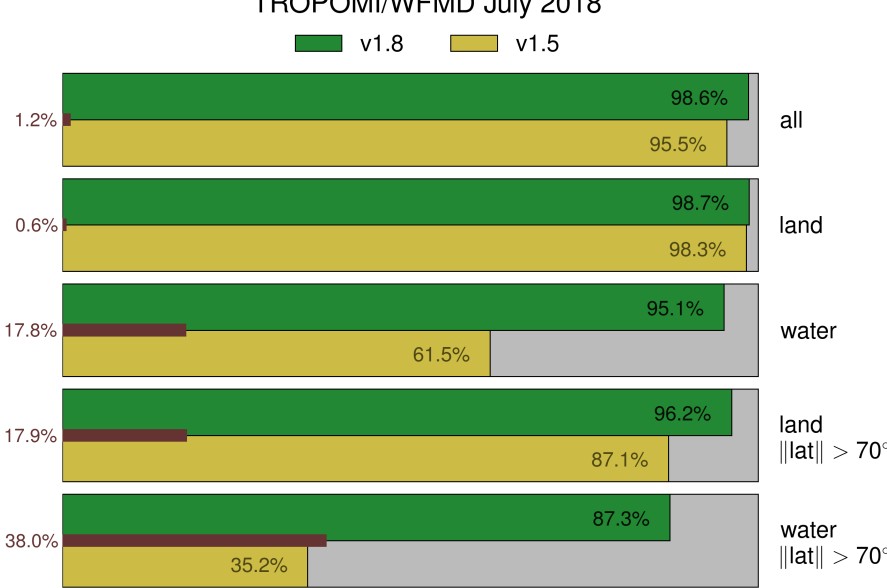

**Figure 8.** Percentage of actually cloud-free scenes according to VIIRS among all scenes passing the quality filter for July 2018 shown in green and yellow for v1.8 and v1.5, respectively. The corresponding relative data loss of good measurements passing the quality filter due to the more stringent filtering in v1.8 compared to v1.5 is highlighted by the red bars.

In v1.8, the quality filter is further improved using 18 million additional (bad quality) ocean scenes equally distributed over the 30 randomly chosen days when training the random forest classifier. This reduces scenes with residual cloud cover especially concerning overcast conditions over the Arctic ocean in summer, which were not always reliably detected in previous product versions. The improvement is demonstrated by the example of July 2018 in Figure 8 showing the precision of the quality filter for cloud-free retrievals. The better the precision, the lower the false discovery rate of class 0, i.e. the fraction of scenes that are classified as good but should in fact be discarded according to VIIRS cloud information. Overall, the percentage of actually cloud-free scenes (defined via the condition that the fraction of VIIRS subscenes classified as *confidently cloudy* has to be smaller than $< 0.1$) in the quality filtered product is already very high in v1.5 ($> 95\%$). However, if restricted to the small subset of measurements at high latitudes over water, it turns out that a significant portion of measurements is misclassified. As this subset only accounts for about $1\%$ of the total measurements, the eminent overall performance of the quality filter is not significantly affected. But for analyses that focus exclusively on high latitudes, the high false discovery rate for this small subgroup may adversely affect conclusive results, e.g. concerning Arctic methane trends. For this reason, the quality filter has been specifically made more stringent in v1.8 so that cloudiness is also detected with sufficient reliability for this challenging subgroup. Figure 9 demonstrates on an example day how the rare cases of cloudy scenes passing the quality filter over the Arctic ocean still visible in v1.5 virtually disappear in v1.8.

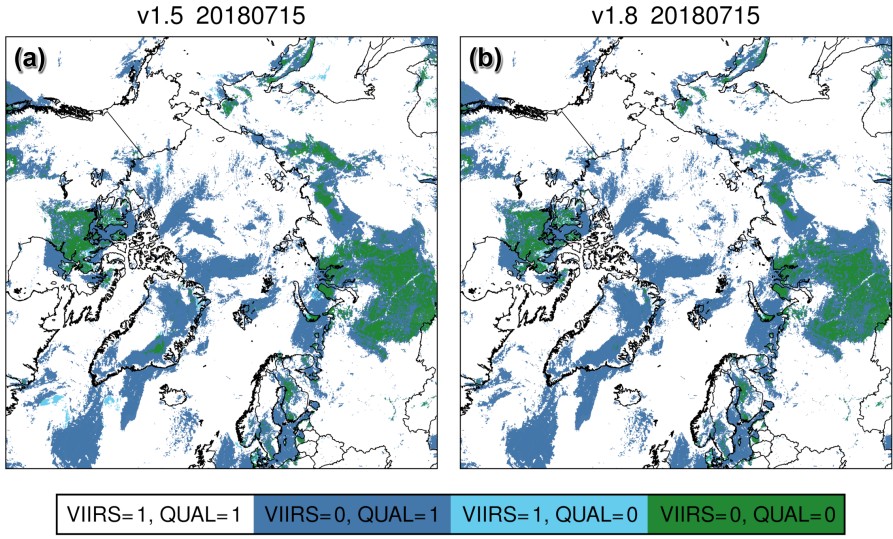

**Figure 9.** Comparison of the implemented quality filter (QUAL, 1: excluded) with the VIIRS cloud classification (1: cloudy) over the Arctic for **(a)** v1.5 and **(b)** v1.8. Matching classifications are shown in white and green. The quality filter is generally stricter than the VIIRS cloud flag (additionally excluded scenes in blue), e.g. for measurements with extremely low albedo such as ocean scenes beyond sun-glint. The rare instances of measurements classified as cloudy by VIIRS but still passing the quality filter (cyan) over the Arctic ocean in v1.5 virtually disappear in v1.8.

The implemented tightening of the quality filter not only increases the precision of class 0, it also decreases the associated recall, i.e the percentage of all the good measurements that correctly pass the quality filter. The corresponding relative data loss is measured by the complement of the ratio of the respective recalls (v1.8 relative to v1.5) and is also shown in Figure 8. The largest loss of good data is observed for the Arctic Ocean (where the precison increase is also largest) with recall being about 40% lower for the updated v1.8 compared to v1.5. It was taken into account that not all cloud-free scenes necessarily result in good retrievals and that the quality filter has also learned to additionally exclude retrievals with obvious methane biases (without using the primary retrieval parameters $XCH_4$ and XCO as features in the prediction of the quality) (Schneising et al., 2019), e.g. measurements with extremely low albedo such as ocean scenes beyond sun-glint or glitter. The trade-off between precision and recall is well justified because precision is much more important than recall in the presented setup. In other words, measurements predicted to be good that should actually be excluded are more critical than cloud-free scenes that are excluded. In the first case, likely biased retrievals are preserved, while in the second case potentially good data is lost, but data quality is not negatively affected for the proportion that passes the filter.

### 3.3.2 Machine learning calibration for methane

Even after quality filtering some systematic errors may still be present in the remaining data due to approximations in the forward model or instrumental issues. As experience has shown, issues related to the surface reflectance and its spectral vari-

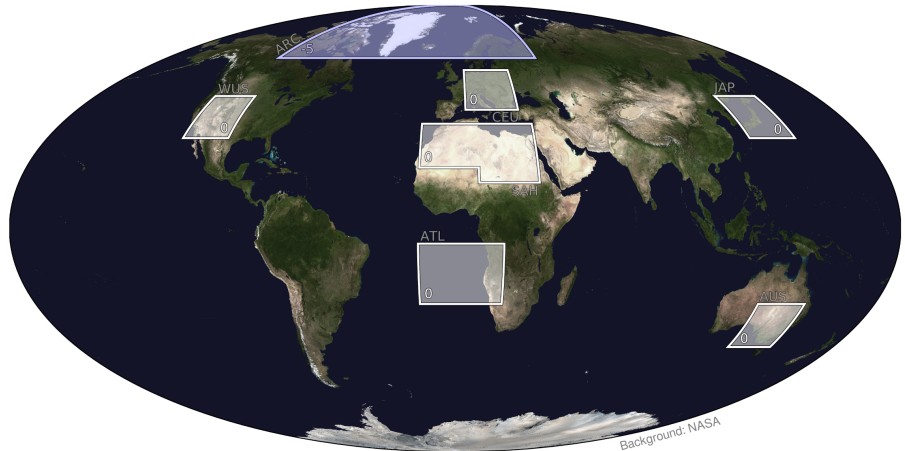

**Figure 10.** Illustration of the updated regions used in the training of the machine learning regressor, namely Arctic (ARC), Western United States (WUS), Central Europe (CEU), Japan (JAP), Sahara (SAH), South Atlantic (ATL), and Australia (AUS). Also given are the respective corrections in ppb for each region applied to the low-resolution methane climatology prior to learning.

ability are of particular concern. Thereby, one has to distinguish between two fundamentally different issues: 1) Biases due to low signals (likely an instrumental issue as the significantly different RemoTeC algorithm shows very similar behaviour), which were addressed by the calibration of the TROPOMI/WFMD algorithm from the start and 2) issues due to the variability

of spectral albedo within the fitting window (independent of the magnitude of the signal). To address this second order effect at least partially, the fitted polynomial coefficient of degree zero $p_0$ is fed into the random forest regressor machine learning calibration as an additional feature since v1.5. This increases the number of features to 6, which are subsequently listed in order of importance: retrieved apparent albedo, across-track dimension index, solar zenith angle, polynomial coefficient $p_0$, cloud parameter $r_{cld}$ (ratio of measured to reference radiance for selected strong $H_2O$ absorption lines as described in Schneising

et al. (2019)), and strong $H_2O$ absorption radiance. As the shallow calibration is not always sufficient to resolve all issues of type 2), the third-degree baseline fit described in Section 3.1 is needed in addition to resolve rare issues with very specific surface types such as the Etosha pan.

Moreover, the training regions and calibration offsets are updated in accordance with Figure 10 to further improve global representativity. The regions are selected to cover the whole range of albedo values and all possible viewing geometries, as

these are important features used in the calibration as listed above. Furthermore, the tree growing is extended to 5000 leaves since v1.5. Up to and including v1.5, the training dataset was limited to scenes with solar zenith angles smaller than $75°$. In v1.8, the learning cut-off threshold is raised to $80°$ to get a better correction for high latitudes. Although demanding conditions with solar zenith angles of more than $75°$ are still excluded in the standard product as they may be subject to scattering and saturation related issues, this yields the possibility to analyse such scenes in an experimental setup if needed. Due to this

learning augmentation, the tree growth was further increased to 10000 leaf nodes. However, compared to the total of about 70 million harvested scenes, the depth of the involved decision trees is still considered shallow. It was explicitly confirmed

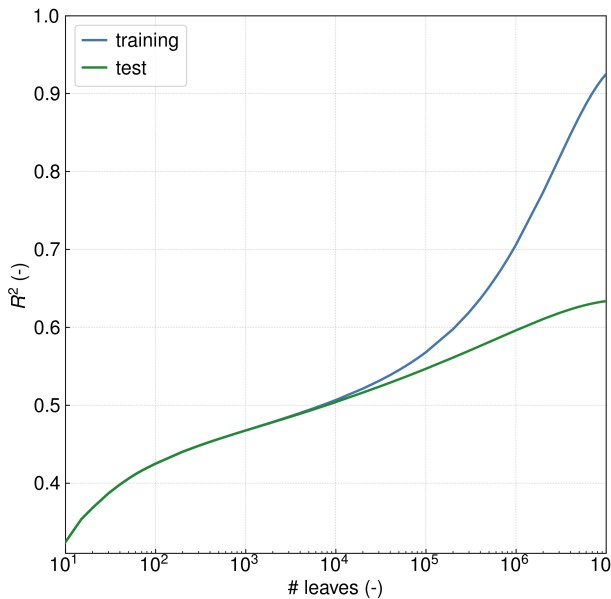

**Figure 11.** Coefficient of determination $R^2$ of the prediction as function of the number of leaves of the decision trees in the random forest calibration. Since $R^2$ of the test dataset increases monotonically with the number of leaves, there is no overfitting with regard to the depth of the decision trees in the random forest. The sweet spot of shallowness is considered to be the number of leaves at which the prediction score functions of training and test data begin to diverge, i.e. about 10000 leaves, which is used as the limit of tree growth in v1.8.

by recursive feature elimination with cross-validation and reservation of $20\%$ of the data for testing purposes that there is no overfitting with regard to the number of features or the depth of the decision trees in the random forest, which is demonstrated in Figure 11. This figure also motivates the choice of the limit of tree growth that was implemented in v1.8, namely the maximum of 10000 leaves.

The calibration is performed against the Simple cLImatological Model for atmospheric $CH_4$ (SLIMCH4) v2021 in nearest neighbour mode without interpolation, whose validation with the Total Carbon Column Observing Network (TCCON) (Wunch et al., 2011) confirmed good quality and indicated that the greatest discrepancies occur in the Arctic (Noël et al., 2022). The biases at the Arctic sites are consistently positive ranging from about $10\,\text{ppb}$ to $20\,\text{ppb}$. Therefore, the values of the climatology in the Arctic region are slightly modified by subtracting a compromise value of $5\,\text{ppb}$ before learning to account for the systematic positive bias in the Arctic, while avoiding potential overcompensation, since typical biases at other sites are in the range of about $\pm 5\,\text{ppb}$. TCCON measurements are only used as a validation source for the climatology, the calibration is otherwise independent of TCCON. Due to the proximity of the regions to TCCON sites used in the validation of the climatology, the validity of the low-resolution estimate of the true atmospheric state is ensured allowing the best possible identification and correction of the coarse-scale interrelationships with other parameters.

The climatology is only used as a coarse approximation of the background to determine the statistical discrepancies regarding the considered (mainly albedo-related) features. After learning of the respective statistical relationships on a temporally and

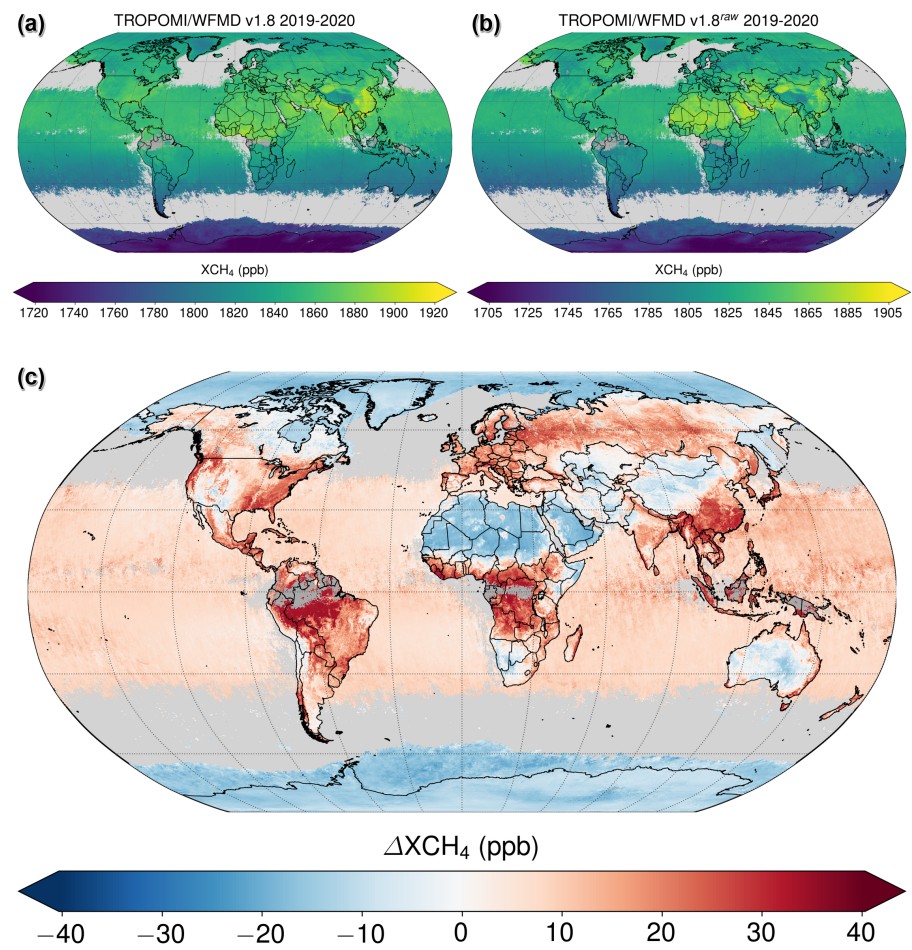

**Figure 12.** Global $XCH_4$ data **(a)** after and **(b)** before calibration. **(c)** shows the calibration $\Delta$ (defined via $XCH_4 = XCH_4^{raw} + \Delta$) after correcting for the global mean difference of $15\,\mathrm{ppb}$.

spatially limited dataset in training (July 2018 until end of 2019 within the regions of Figure 10), the climatology is not explicitly used anymore and the calibration is performed only using the 6 intrinsic parameters of the retrieval listed above, in particular the retrieved apparent albedo. As a consequence, the corresponding calibration shown globally in Figure 12 and seasonally resolved in Figure 13 mainly reflects albedo features. The average global correction amounts to $(15\pm12)\,\mathrm{ppb}\,(1\sigma)$, which is similar in magnitude to the correction applied in the operational/RemoTeC product (Lorente et al., 2021), although the detailed spatial patterns of the correction are somewhat different. The corresponding standard deviation of the correction is well below the natural methane variability on a global scale, e.g. due to latitudinal gradients.

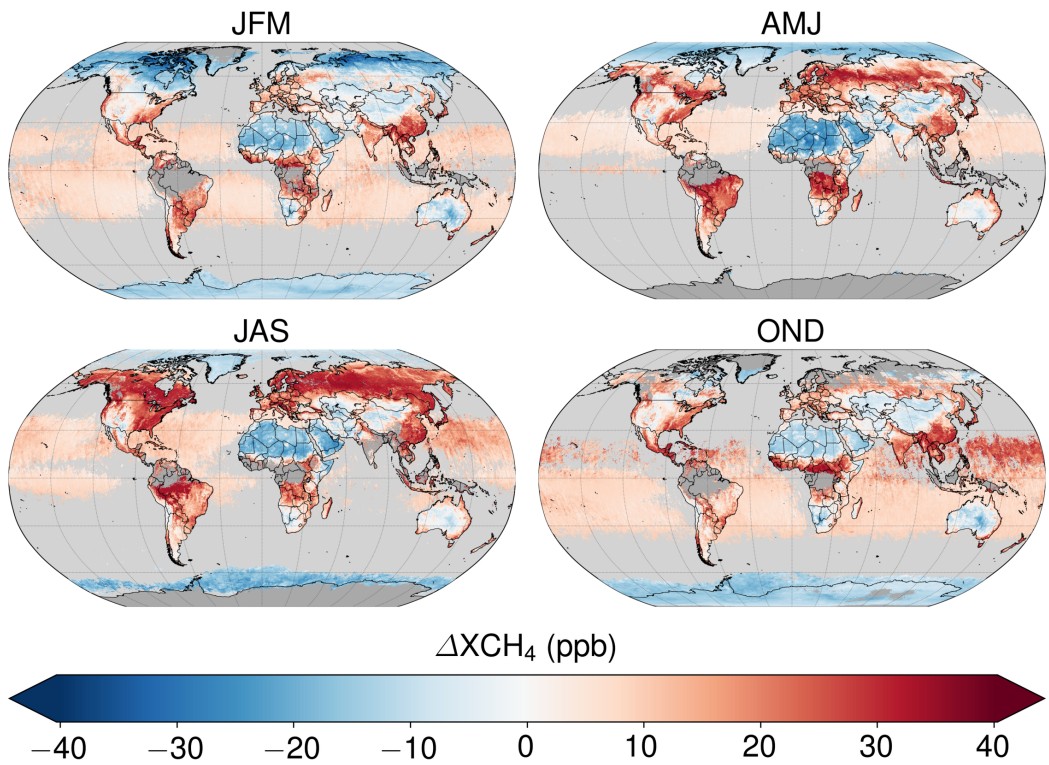

**Figure 13.** The global calibration as in Figure 12 but for the different seasons.

## 3.4 Wavelet decomposition and destriping

There is striping in flight direction in the TROPOMI $XCH_4$ and XCO data presumably due to different offsets and gains of pixels of the detector array used for the staring pushbroom concept (Borsdorff et al., 2019; Schneising et al., 2019). The inclusion of the across-track index in the calibration accounts for recurring systematics, e.g. potential smooth biases induced by viewing zenith angle or temporally constant striping patterns. As a result, striping is improved to some extent but complete destriping, in particular with respect to temporally variable striping patterns, cannot be entirely achieved by a shallow calibration. Therefore, the remaining vertical stripes in the satellite data after calibration are efficiently removed orbitwise in v1.8 by combined wavelet-Fourier filtering while optimally preserving all other spatial information (Münch et al., 2009). The basic procedure for gapless data is described in the following subsection.

### 3.4.1 Basic procedure

First a 2D multilevel wavelet decomposition with symmetric boundary conditions is performed orbitwise separating horizontal, vertical, and diagonal details. More specifically, a wavelet decomposition of level $L$ divides the 2D signal $f(x, y)$ into a low-frequency approximation (represented by a scaling function $\Phi_L$ and coefficients $a_L$), which still contains the self-similar

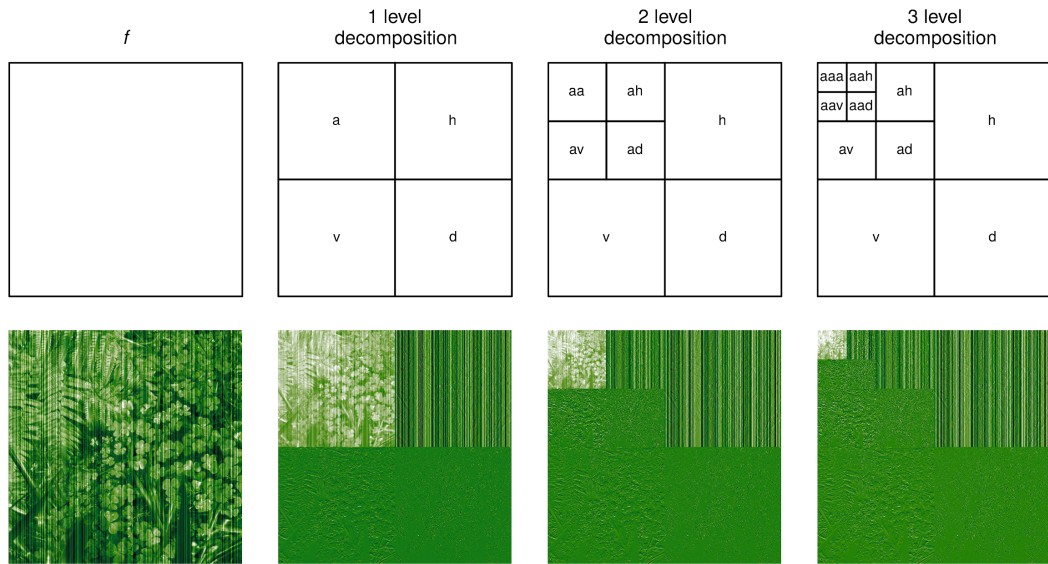

**Figure 14.** 2D wavelet decomposition of an image with vertical stripes for different decomposition levels. The low-pass approximation coefficients $a_l$ still contain the self-similar complete signal information at a lower resolution; the other bands contain the details to reconstruct the higher resolution image. The striping information is concentrated to the detail bands represented by the coefficients $h_l$.

complete signal information at lower resolution, and high-frequency detail bands (represented by wavelet functions $\Psi_l$ and coefficients $h_l$, $v_l$, $d_l$ for different scales $l \in \{1, \ldots, L\}$). The set of coefficients representing the wavelet representation $\mathcal{W}$ of $f(x, y)$ allows the lossless reconstruction of the original information:

$$f(x,y) = \sum_{m,n} a_{L,m,n} \Phi_{L,m,n}(x,y) + \sum_{b \in \{h,v,d\}} \sum_{l=1}^{L} \sum_{m,n} b_{l,m,n} \Psi_{b,l,m,n}(x,y) \tag{2}$$

$$f(x,y) \Longleftrightarrow \mathcal{W} = \left\{ a_L, h_l, v_l, d_l \,\middle|\, l \in \{1, \ldots, L\} \right\} \tag{3}$$

The 2D wavelet decomposition of an image with vertical stripes is demonstrated in Figure 14. The striping information is concentrated in the corresponding horizontal detail bands $h_l$.

In the following, only the fraction of the detail bands affected by stripes are Fourier-filtered while the other bands remain unchanged. Using fast Fourier transformation (FFT), the stripe information in the affected detail bands $h_l$ is further condensed around $Y = 0$ in the frequency domain and attenuated by multiplication of the Fourier coefficients with a Gaussian function

$$g(X,Y) = 1 - \exp\left( \frac{-Y^2}{2\sigma^2} \right) \tag{4}$$

In the case of imperfect stripes with offset variability in stripe direction, the attenuation parameter $\sigma$ has to be chosen larger than for ideal stripes to enable efficient destriping. However, $\sigma$ should be as small as possible because original image information is also increasingly removed when $\sigma$ becomes too large. The optimal choice of $\sigma$ depends on the image and the characteristics of the striping artifacts.

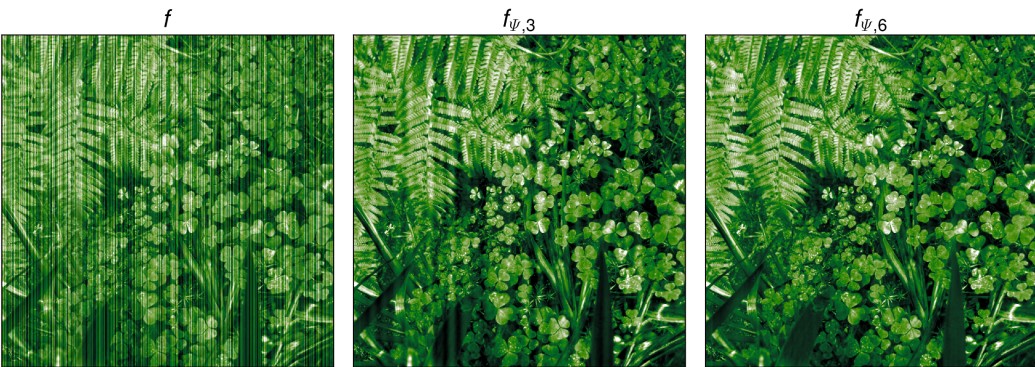

**Figure 15.** Performance of the combined wavelet-Fourier filter using coiflet wavelets (coif16), $\sigma = 2$, and different decomposition levels $L$.

Attenuation with $g(X, Y)$ and subsequent inverse FFT results in updated wavelet coefficients $\tilde{h}_l$. Reconstruction using these refined coefficients together with the original coefficients for the other bands results in the destriped signal $f_{\Psi,L}(x, y)$:

$$f_{\Psi,L}(x,y) \Longleftrightarrow \tilde{\mathcal{W}} = \left\{ a_L, \tilde{h}_l, v_l, d_l \middle| l \in \{1, \ldots, L\} \right\} \tag{5}$$

Combined wavelet-Fourier filtering has a better destriping performance than Fourier filtering alone because the majority of the coefficients remains entirely unchanged ensuring that the structural features outside the detail bands affected by stripes are preserved (Münch et al., 2009). The destriping performance is demonstrated in Figure 15 using coiflets, which are near symmetric compactly supported wavelets that are useful in signal processing due to the high number of vanishing moments for both the scaling and the wavelet functions (Monzón et al., 1999). The scaling and wavelet functions of coiflets are illustrated in Figure A1.

### 3.4.2 Application to satellite data

Before the combined wavelet-Fourier destriping algorithm can be applied to satellite orbits $f(x, y)$, data missing from quality filtering must be filled in suitably. Initially, rows without data at all are filled with the median of $f$. For rows with data, the gaps in each row are filled with the row-wise median value and a stripe function $s_y(x)$ is determined by subtracting the median value from the original row $y$. A fitted cubic polynomial is also subtracted on the support of the original data to remove smooth horizontal gradients of $s_y$. Afterwards, the column-wise median of the previously calculated stripe functions is computed for all columns to add the median striping to the filled gaps. This results in $F(x, y)$ with completely filled domain and extended stripes in vertical direction meeting the requirements for wavelet-Fourier filtering. After destriping of $F$, the function is restricted to the original support. The whole procedure

$$f \longrightarrow F \longrightarrow F_\Psi \longrightarrow f_\Psi = F_\Psi|_{\mathrm{supp}(f)} \tag{6}$$

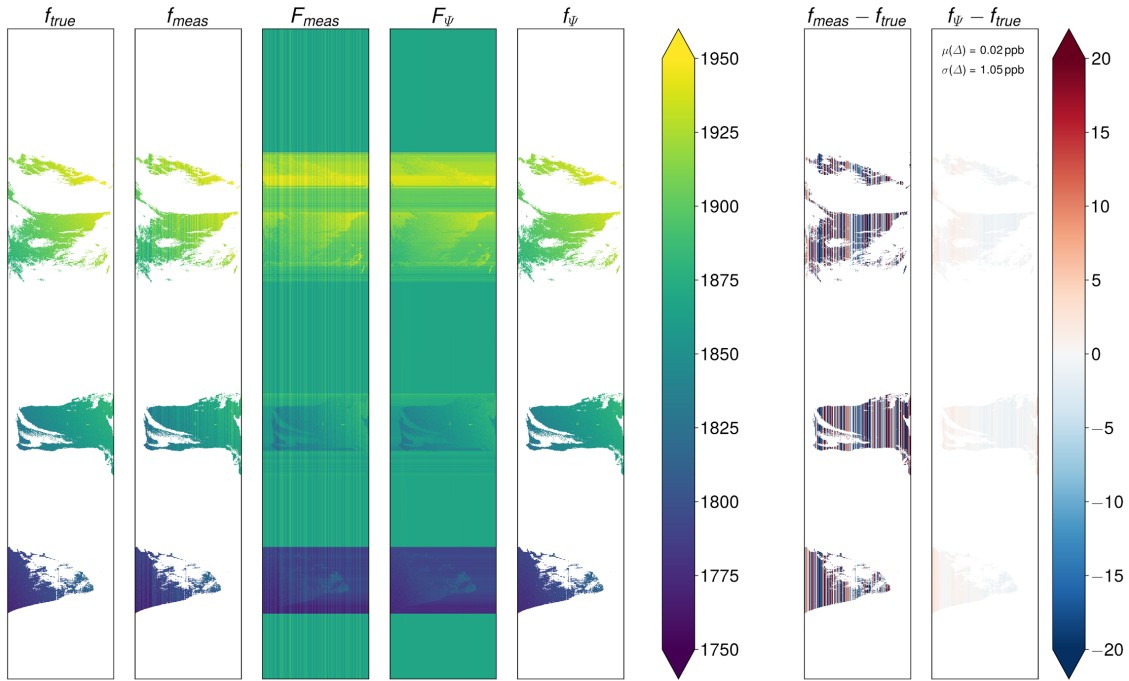

**Figure 16.** Illustration of the performance of combined wavelet-Fourier filtering for a synthetic orbit with significant gaps using Coiflets (coif16), $\sigma = 2$, and $L = 7$. Thereby, $f_{\mathrm{true}}$ is the true distribution, $f_{\mathrm{meas}}$ is the distribution as measured by the instrument (with stripes), $F_{\mathrm{meas}}$ is the distribution after filling, $F_\Psi$ is the filter result of $F_{\mathrm{meas}}$, and $f_\Psi$ is the final destriped result after restriction to the original support. The other two columns show the respective differences of the measured and destriped distribution to the truth.

is illustrated in Figure 16 demonstrating that the original true distribution is very well reconstructed by the destriping algorithm. In the case of the TROPOMI/WFMD products, coiflets of order 16 (coif16), $\sigma = 2$, and $L = 7$ proved to be suitable choices for the wavelet family, the attenuation parameter, and the wavelet decomposition level.

The performance of the destriping filter is shown in Figures 17 and 18 for genuine TROPOMI data using two example scenes in Turkmenistan and the United States known for large methane emissions from the oil and gas industry and two example scenes in India and Central Europe with carbon monoxide emissions from the steel industry. Just as in Figures 4 and 5 for the Etosha pan, it is immediately apparent that the striping decreases significantly in the latest product version for both $XCH_4$ and XCO. For comparison, the figures also include the operational products, which also exhibit striping artefacts. As the operational
carbon monoxide product only includes total columns, XCO is computed using dry air columns obtained from the ECMWF ERA5 reanalysis as for TROPOMI/WFMD. A destriping approach for a pre-operational version of the operational TROPOMI CO retrieval was introduced (Borsdorff et al., 2019) and implemented in the operational processing of offline (OFFL) data for orbits since July 2021.

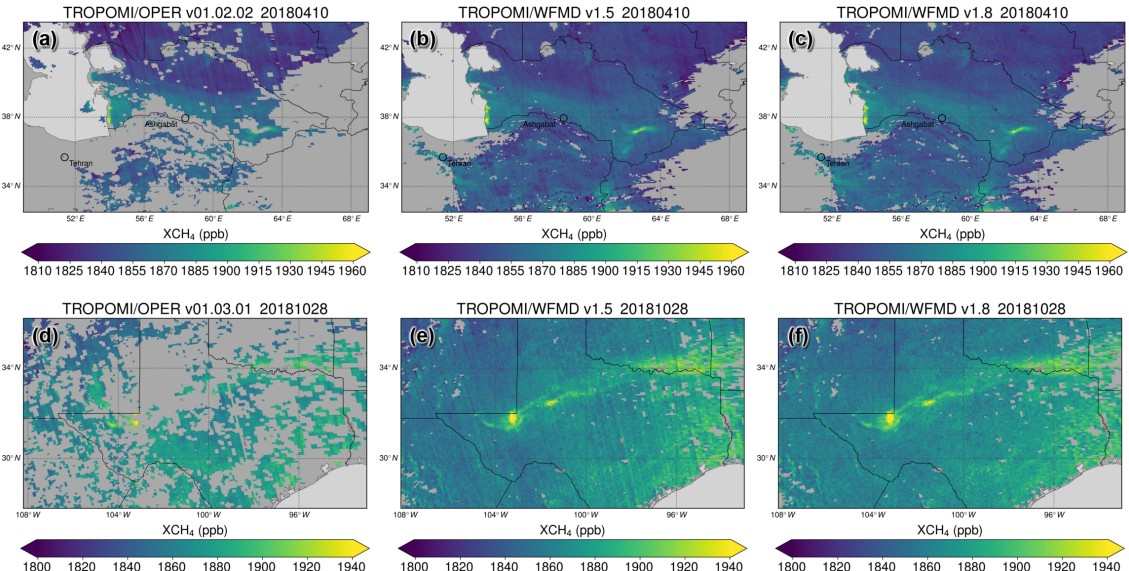

**Figure 17.** Performance of the destriping filter for the $XCH_4$ product for two example scenes in **(a-c)** Turkmenistan and **(d-f)** the United States with enhancements due to emissions from the oil and gas industry. Panels **(b)** and **(e)** show the previous WFMD version, panels **(c)** and **(f)** the improved version with destriping. The operational product also exhibiting striping is shown in panels **(a)** and **(d)** for comparison.

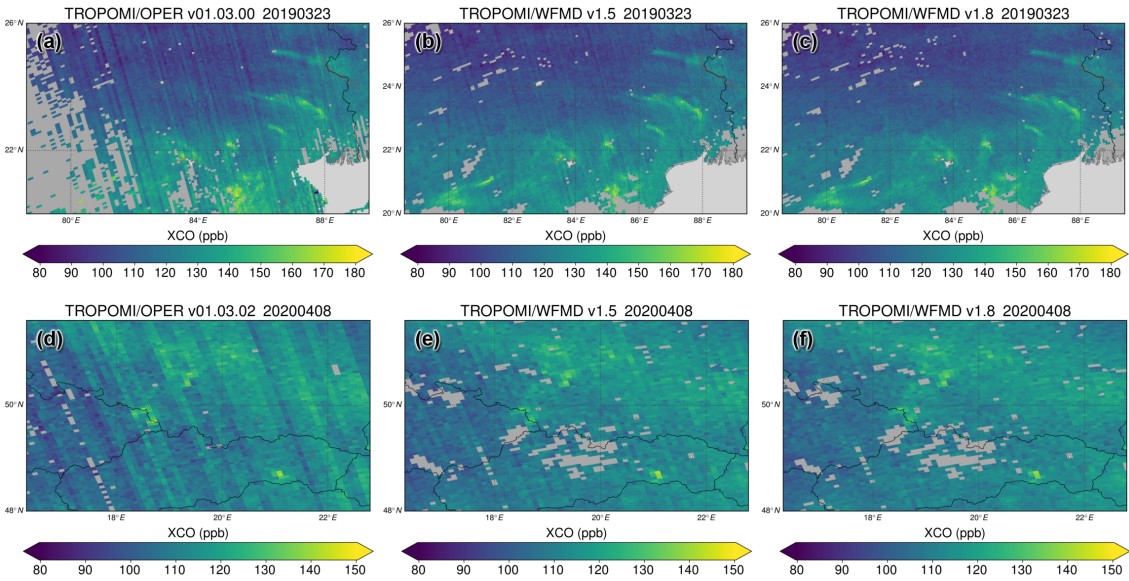

**Figure 18.** Performance of the destriping filter for the XCO product for two example scenes in India **(a-c)** and Central Europe **(d-f)** with enhancements due to emissions from the steel industry. Panels **(b)** and **(e)** show the previous WFMD version, panels **(c)** and **(f)** the improved version with destriping. The operational product also exhibiting striping is shown in panels **(a)** and **(d)** for comparison.

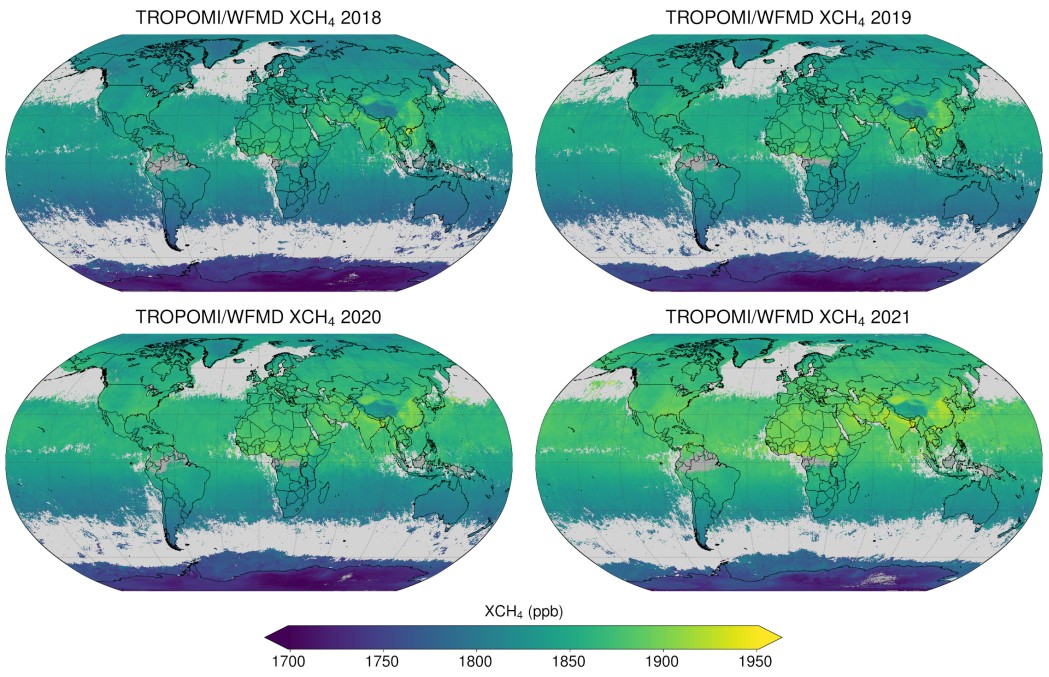

**Figure 19.** Yearly averages of TROPOMI/WFMD v1.8 XCH$_4$ shown on a $0.1° \times 0.1°$ grid.

## 4 Overview of the products

The improved XCH$_4$ and XCO products currently cover a period of 4.5 years, starting at mission launch in autumn 2017 up
to and including April 2022, and will be regularly updated. Global yearly distributions from 2018 to 2021 of both products
are shown in Figures 19 and 20, respectively. As can be seen, the quality filtered datasets have similar coverage from year to
year including high latitudes and the oceans. For both gases the interhemispheric gradient and increments over major source
regions, such as those related to anthropogenic emissions in China, India, and Southeast Asia, are clearly detected. In the XCO

case, source regions associated with biomass burning (tropical Africa and South America) and wildfires (Canada and Siberia)
are also visible. While there is an accelerating growth of atmospheric methane for the considered time period, the annual mean
increase of XCO is decelerating with almost stable mean values in 2020 and 2021, which are slightly larger than in the two
previous years, presumably due to stronger wildfire activity in the boreal zone.

    To further investigate the annual increase of the products, time series of globally-averaged monthly gridded data ($0.1° \times 0.1°$,

weighted by area) are decomposed into a seasonal and a trend component using Locally Estimated Scatterplot Smoothing
(LOESS) (Cleveland et al., 1990). LOESS smoothes a function of variables by local polynomial regression using a smoothing
kernel based on the tricube weight function with compact support. The kernel width can be adjusted separately for the different
components by specifying related smoothing parameters. To get a decomposition that is robust to outliers, two recursive
procedures are performed: an inner loop updating the seasonal and trend components nested inside an outer loop computing

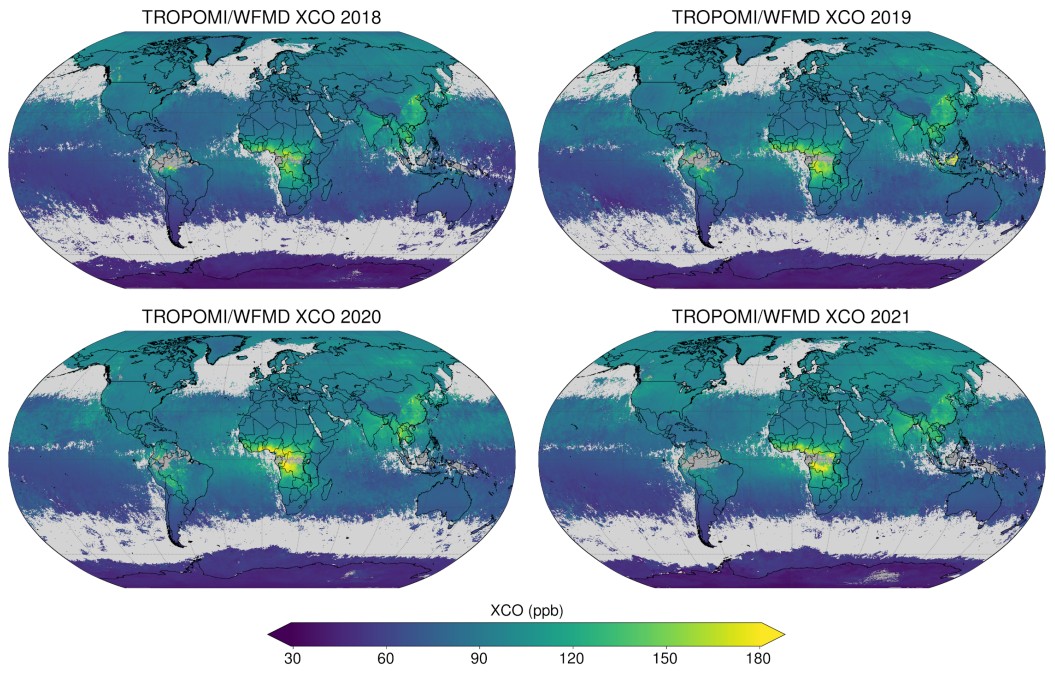

**Figure 20.** As Figure 19 but for TROPOMI/WFMD v1.8 XCO.

robustness weights to be used in the next run of the inner loop. The Seasonal and Trend decomposition using LOESS (STL) has several advantages over simpler separation methods such as using a 12-month running mean: 1) the trend is computed for the entire data record including both ends of the time series, 2) the smoothness of the components can be controlled by the user, 3) the component estimates are not affected by occasional outliers in the robust estimation. STL has 6 different parameters to specify: the periodicity of the sequence $n_p$, the smoothing parameters $n_l$, $n_t$, $n_s$ for low-pass filter, trend, and seasonal component, as well as the number of passes through the inner and the outer loop, $n_i$ and $n_o$. There is also a generalisation called MSTL to support multiple seasonalities by applying the STL procedure iteratively to seamlessly separate the different seasonal cycles (Bandara et al., 2021).

We follow the original implementation of Cleveland et al. (1990) using $n_i = 5$ and $n_o = 0$ as the robust estimation of seasonal and trend components is not needed, $n_p = 12$ for monthly data, $n_l = \lceil n_p \rceil_{odd} = 13$, and $n_t = \lceil 1.5 \cdot n_p / (1 - 1.5/n_s) \rceil_{odd}$. According to this settings the only free parameter is $n_s$, which then also determines $n_t$ by the formula given above. We choose $n_s = 27$ as compromise between the mutually dependent smoothnesses of the seasonal and the trend component. We restrict this analysis to data since April 2018, because the data density during the previous commissioning phase of TROPOMI is significantly reduced. Empirical results from both marine surface and TROPOMI satellite data have shown that the residual component decreases significantly when adding an extra seasonality $n_{\tilde{p}} = 9$ to the decomposition. Therefore, we perform a corresponding MSTL with 100 iterations to separate the two seasonalities $n_p$ and $n_{\tilde{p}}$. After decomposition, the annual increase

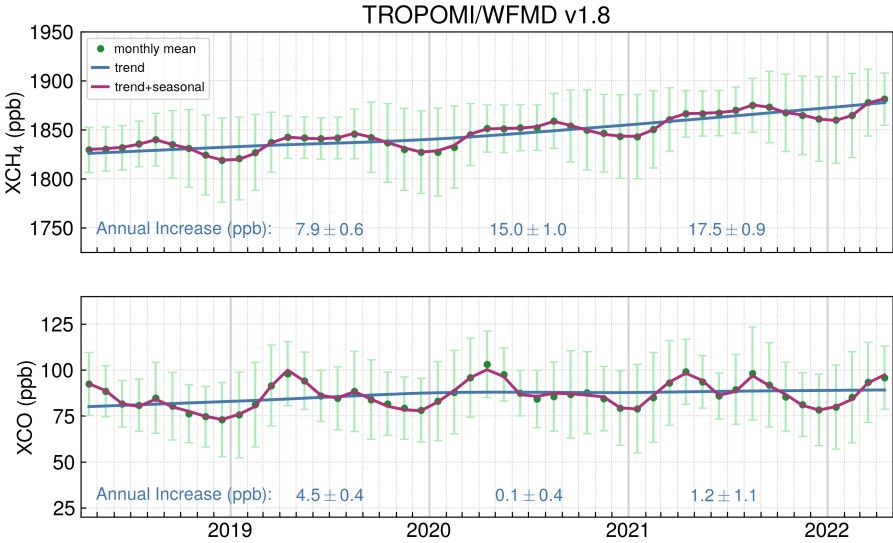

**Figure 21.** Time series of globally-averaged monthly gridded TROPOMI/WFMD v1.8 data. The underlying area-weighted monthly means are shown as green dots with the associated standard deviation as bars in lighter green. The season-trend decomposition used to derive the annual increase is done using Locally Estimated Scatterplot Smoothing (LOESS). The trend component is shown in blue and the sum of trend and seasonal component is shown in purple.

in a given year is defined as the increase in the abundance of the trend component from the January mean value in that year to the January mean value of the next year. The monthly means and the components are shown in Figure 21.

As TROPOMI is a passive spectrometer, the latitudinal coverage of quality filtered data differs with season as a consequence of the different illumination conditions, e.g. there is no good data at high latitudes during polar night. Although these regional
seasonal gaps may limit the independent trend analysis for small latitude bands in polar regions, the presented analysis of annual increases based on globally-averaged monthly data is not severely affected because there are no gaps in the global time series and the seasonal sampling of the temporally consistent TROPOMI/WFMD products is similar from year to year. The impact of the choice of the seasonal smoothing parameter $n_s$, the extra seasonality $n_{\tilde{p}}$, and the different seasonal sampling is taken into account in the uncertainty analysis.
The resulting annual increases corroborate the accelerated growth of $XCH_4$ (2019: $7.9\,\text{ppb}$, 2020: $15.0\,\text{ppb}$, 2021: $17.5\,\text{ppb}$) and the decelerated growth of XCO (2019: $4.5\,\text{ppb}$, 2020: $0.1\,\text{ppb}$, 2021: $1.2\,\text{ppb}$) during the covered period already indicated in the annual averages of Figures 19 and 20. The uncertainties of the annual increases are estimated by two times the root sum square of four components: the standard deviation of a set of increases induced by 1) bootstrap resampling of the global gridboxes contributing to the monthly means, 2) randomly modifying the data to take the uncertainties of the retrieved
column-averaged mole fractions into account, 3) randomly modifying the seasonal smoothing parameter $n_s$ as well as randomly including or excluding high latitudes ($\|\text{lat}\| > 70°$) and the additional seasonality $n_{\tilde{p}}$ independently of each other, and

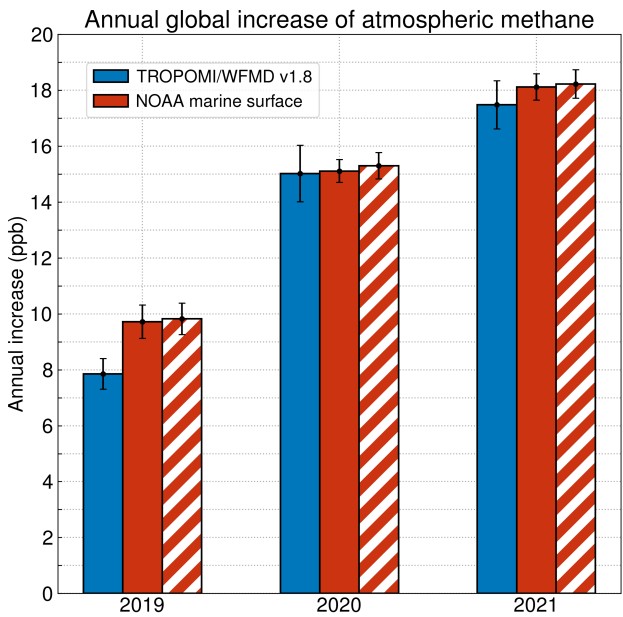

**Figure 22.** Comparison of the annual globally-averaged increases of TROPOMI/WFMD v1.8 XCH$_4$ (blue) and NOAA marine surface CH$_4$ (red) as reported by Lan et al. (2022). The hatched red bar is a consistency check illustrating the results when using the method presented here to determine the increases on the globally-averaged monthly means provided by NOAA (Lan et al., 2022).

4) randomly shifting the timeseries by 0 or $\pm 1$ time steps. The corresponding sample size in the estimation of all uncertainty components is 5000, i.e. 5000 alternative increases are calculated per year and component to be included in the analysis.

As can be seen from the comparison in Figure 22, the derived temporal development of the annual global methane increases is consistent with the trends determined from marine surface sites of the Global Monitoring Division of NOAA's Earth System Research Laboratory (Lan et al., 2022) with two consecutive annual record growth rates in 2020 and 2021.

To compare the performance of the different TROPOMI/WFMD versions, the products are validated with the ground-based Total Carbon Column Observing Network (TCCON) (Wunch et al., 2011), which uses similar Fourier Transform Spectrometer (FTS) instrumentation and a common retrieval algorithm network-wide. Table 2 compares the most important figures of merit obtained by validation with the GGG2014 version of TCCON data at 26 sites (see Table B1) using the well-established validation procedure described in detail in Schneising et al. (2019). The values for v1.2 are somewhat different from the results in Schneising et al. (2019) because there are 5 additional TCCON sites involved in the present validation and the validation time period is also extended by 1 year. In addition to the spatial systematic error, defined as the the standard deviation of the local offsets relative to TCCON at the individual sites, the seasonal systematic error is also taken into account, which is defined as the standard deviation of the four overall seasonal offsets using all TCCON sites combined. The total (spatio-temporal) systematic error is then determined by the root-sum-square of the spatial and seasonal systematic errors. We use the TCCON

**Table 2.** Comparisons of the most important figures of merit for the different TROPOMI/WFMD product versions obtained by validation with the GGG2014 version of TCCON data at 26 sites (see Table B1). The random error is measured by the scatter of the satellite data relative to the TCCON, the systematic error is a combination of the spatial station-to-station bias and the seasonal bias. $N$ is the total number of collocations for all sites together.

| Data product | Random error (ppb) | | | Systematic error (ppb) | | | $N$ | | |
|---|---|---|---|---|---|---|---|---|---|
| | v1.2 | v1.5 | v1.8 | v1.2 | v1.5 | v1.8 | v1.2 | v1.5 | v1.8 |
| Methane ($XCH_4$) | 14.2 | 12.8 | 12.4 | 5.0 | 5.1 | 5.2 | 615435 | 649041 | 614469 |
| Carbon monoxide (XCO) | 5.3 | 5.3 | 5.1 | 2.4 | 2.3 | 2.6 | | | |

GGG2014 version instead of the newer GGG2020 product, because not all sites have been reprocessed at time of submission. The analysis of regional biases based on GGG2014 is therefore more robust due to the larger number of regions covered.

To ensure comparability of the results for the different versions, the comparison was restricted consistently to a match period ending 2020. Since the derived systematic errors of the satellite products are comparable in magnitude to the station-to-station $1\sigma$-accuracy of the TCCON, estimated to be $2\,\mathrm{ppb}$ for XCO and $3.5\,\mathrm{ppb}$ for $XCH_4$ (Wunch et al., 2010), all TROPOMI/WFMD products are equivalent in terms of systematic error, although there are small deviations (in the range of tenths of ppb) for the different versions. Since the precision of TCCON is appreciably better compared to the satellite data, the improvement in the random error is considered significant, in particular in the case of methane, and is ascribed to the reduction of the pseudo-noise component due to the algorithm improvements, e.g. the efficient destriping algorithm and the optimised quality filter reducing the number of outliers.

## 5 Conclusions

We have introduced the changes implemented in the latest version v1.8 of the combined scientific TROPOMI/WFMD $XCH_4$ and XCO product, which currently covers 4.5 years of data from mission start until end of April 2022 and will be further extended in the future. It was demonstrated that the performance of the updated retrieval algorithm was further improved, for example with respect to striping in flight direction for single overpasses due to a dedicated destriping algorithm, which simultaneously preserves the actual spatial trace gas features. Together with the other advancements, such as an optimised quality filter reducing the number of outliers and the usage of an improved Digital Elevation Model, this reduces the pseudo-noise component resulting in an improved random error estimated by validation with the Total Carbon Column Observing Network.

Due to the machine learning-based quality filter, current cloud information is no longer required after the initial supervised learning process is completed. As a consequence, the algorithm is independent of the availability or consistency of cloud products, such as VIIRS, as the algorithm has learned to classify good quality measurements entirely by intrinsic parameters available from or used in the preceding processing excluding the primary retrieval parameters $XCH_4$ and XCO. The resulting

consistent data products with similar spatial coverage from year to year are not only generally useful for detecting and quantifying emission sources, but also enable long-term applications such as trend determination. In the case of methane, the derived global annual increases are consistent with the trends determined from marine surface sites of the Global Monitoring Division of NOAA's Earth System Research Laboratory exhibiting an accelerated growth for the period covered by the TROPOMI data with two consecutive annual record growth rates in 2020 and 2021.

Although the natural and anthropogenic source and sink processes for $CH_4$ and CO are known, it is still a challenge to identify the exact contributions of the different processes to the observed annual growth rate variations. Especially for methane this lack of detailed understanding is critical as it complicates climate projections and the specification of effective emission mitigation strategies. A better estimate of the source and sink budget can be inferred by exploiting a comprehensive monitoring system, which compiles complementary information from accurate local in situ measurements and satellite observations of

ample coverage, within an inverse modelling framework. High quality TROPOMI products with their unique combination of high precision, spatiotemporal resolution, and global coverage offer a valuable opportunity in this context.

## Appendix A: Coiflet wavelets

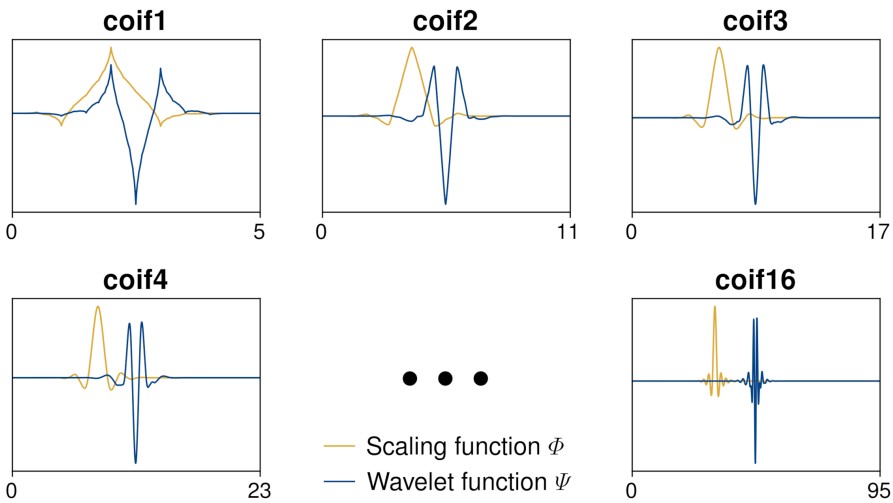

**Figure A1.** Scaling and wavelet functions for the coiflet wavelet family.

## Appendix B: List of TCCON sites

**Table B1.** TCCON sites used in the validation sorted by latitude from north to south.

| Station | Latitude (°) | Longitude (°) | Altitude (km) | Reference |
|---|---|---|---|---|
| Eureka | 80.05 | −86.42 | 0.61 | Strong et al. (2019) |
| Ny-Ålesund | 78.92 | 11.92 | 0.02 | Notholt et al. (2017) |
| Sodankylä | 67.37 | 26.63 | 0.19 | Kivi et al. (2014) |
| East Trout Lake | 54.35 | −104.99 | 0.50 | Wunch et al. (2018) |
| Białystok | 53.23 | 23.03 | 0.19 | Deutscher et al. (2015) |
| Bremen | 53.10 | 8.85 | 0.03 | Notholt et al. (2014) |
| Karlsruhe | 49.10 | 8.44 | 0.11 | Hase et al. (2015) |
| Paris | 48.85 | 2.36 | 0.06 | Te et al. (2014) |
| Orléans | 47.97 | 2.11 | 0.13 | Warneke et al. (2014) |
| Garmisch | 47.48 | 11.06 | 0.75 | Sussmann and Rettinger (2018) |
| Park Falls | 45.94 | −90.27 | 0.44 | Wennberg et al. (2017) |
| Rikubetsu | 43.46 | 143.77 | 0.38 | Morino et al. (2018c) |
| Lamont | 36.60 | −97.49 | 0.32 | Wennberg et al. (2016b) |
| Anmeyondo | 36.54 | 126.33 | 0.03 | Goo et al. (2014) |
| Tsukuba | 36.05 | 140.12 | 0.03 | Morino et al. (2018a) |
| Nicosia | 35.14 | 33.38 | 0.19 | Petri et al. (2020) |
| Edwards | 34.96 | −117.88 | 0.70 | Iraci et al. (2016) |
| JPL | 34.20 | −118.18 | 0.39 | Wennberg et al. (2016a) |
| Caltech | 34.14 | −118.13 | 0.24 | Wennberg et al. (2015) |
| Saga | 33.24 | 130.29 | 0.01 | Shiomi et al. (2014) |
| Burgos | 18.53 | 120.65 | 0.04 | Morino et al. (2018b) |
| Ascension Island | −7.92 | −14.33 | 0.03 | Feist et al. (2014) |
| Darwin | −12.46 | 130.93 | 0.04 | Griffith et al. (2014a) |
| Réunion | −20.90 | 55.49 | 0.09 | De Mazière et al. (2017) |
| Wollongong | −34.41 | 150.88 | 0.03 | Griffith et al. (2014b) |
| Lauder | −45.04 | 169.68 | 0.37 | Sherlock et al. (2014) |

*Data availability.*   The methane and carbon monoxide data products presented in this publication are available at http://www.iup.uni-bremen.
de/carbon_ghg/products/tropomi_wfmd/.

*Author contributions.*   OS designed and operated the TROPOMI/WFMD satellite retrievals, performed the data analysis, interpreted the results, and wrote the paper. MiB, JH, SV, MR, MaB, HB, and JPB provided significant conceptual input to the design of the retrievals and the improvement of the paper. All authors discussed the results and commented on the paper.

*Competing interests.*   The authors declare that they have no conflict of interest.

*Acknowledgements.*   This publication contains modified Copernicus Sentinel data (2018-2022). Sentinel-5 Precursor is an ESA mission implemented on behalf of the European Commission. The TROPOMI payload is a joint development by ESA and the Netherlands Space Office (NSO). The Sentinel-5 Precursor ground-segment development has been funded by ESA and with national contributions from The Netherlands, Germany, and Belgium.

We acknowledge the use of VIIRS imagery from the NASA Worldview application (https://worldview.earthdata.nasa.gov/) operated by the NASA/Goddard Space Flight Center Earth Science Data and Information System (ESDIS) project and thank the European Centre for Medium-Range Weather Forecasts (ECMWF) for providing the ERA5 reanalysis. TCCON data were obtained from the TCCON Data Archive, hosted by CaltechDATA, California Institute of Technology (https://tccondata.org/); we thank the TCCON partners for their efforts to operate the TCCON sites, which are a essential validation resource for satellite products.

The Python packages *scikit-learn* (Pedregosa et al., 2011) and *PyWavelets* (Lee et al., 2019) are integral components in the implementation of the machine-learning-based quality filter/calibration and the wavelet decomposition used in the destriping procedure.

*Financial support.*   The research leading to the presented results received funding from the European Space Agency (ESA) via the projects GHG-CCI+ and MethaneCAMP (ESA contract nos. 4000126450/19/I-NB and 4000137895/22/I-AG) and from the german ministry of education and research (BMBF) within its project ITMS via grant 01 LK2103A. The TROPOMI/WFMD retrievals presented here were performed on HPC facilities of the IUP, University of Bremen, funded under DFG/FUGG grant nos. INST 144/379-1 and INST 144/493-1. The research received additional funding from the University of Bremen as part of the junior research group "Greenhouse gases in the Arctic" and from the Deutsche Forschungsgemeinschaft (DFG project no. 268020496 - TRR 172) within the Transregional Collaborative Research Center "ArctiC Amplification: Climate Relevant Atmospheric and SurfaCe Processes, and Feedback Mechanisms (AC)[3]". The article processing charges for this open-access publication were covered by the University of Bremen.

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
