# Peer review of "Advances in retrieving XCH4 and XCO from Sentinel-5 Precursor: Improvements in the scientific TROPOMI/WFMD algorithm"

_Atmospheric Measurement Techniques, 2022_

## Author Comment (AC1)

**Final response to referee comments on paper amt-2022-258**

First of all, we would like to thank reviewer #2 for his/her constructive comments, which helped to improve the manuscript. In particular, there are 9 additional and several revised Figures, as well as a more detailed description and discussion of the third-order baseline fit and methane calibration. Below we give answers and clarifications to all comments made by the referee (repeated in italics).

**Anonymous Referee #2**

**General comments**

**Reviewer:** *1) The title does not clearly reflect the contents of the paper. This work deals with algorithm updates to TROPOMI/WFMD. It does not introduce new retrieval concepts (the only change to the retrieval approach is an update in the handling of the spectral baseline during fitting). In addition, some algorithm updates proposed in this manuscript are discussed with a strong focus on XCH4 retrievals, some even disregarding XCO. The title should therefore be changed to focus on the content of the paper. I propose "TROPOMI/WFMD XCH4 v1.8: improvements in spectral fitting, auxiliary datasets and post-processing"*

**Authors:** From our point of view, the updates regarding quality filter, calibration, and destriping procedure are also part of the retrieval algorithm. Most updates affect XCO in the same way (with the exception of the machine learning calibration). The focus is on XCH$_4$ as the requirements are more stringent and the presented improvements are thus more relevant to meet them. Instead of exclusively covering XCH$_4$, we will discuss XCO in more detail. The title is changed to "Advances in retrieving XCH$_4$ and XCO from Sentinel-5 Precursor: Improvements in the scientific TROPOMI/WFMD algorithm" to better reflect the contents of the paper. Further details are provided in the abstract and do not need to be listed in the title.

**Reviewer:** *2) I am concerned that the authors do not appropriately cite work that they have presented in previous articles. Two examples stand out to me.*

*Firstly, section 3.2 does not provide significant new information on WFMD v1.8 retrievals, since the algorithm updates related to the digital elevation model have already been discussed in a recent AMT article by the same team (Hachmeister et al. 2022). In fact, Figure 4 of the present article is essentially identical to Figure 10 of Hachmeister et al.. The authors should not publish these results twice. In my opinion, section 3.2 should therefore be removed.*

*Secondly, in line 68, please remove reference Buchwitz et al. 2017. That article is neither about TROPOMI XCH4 nor about WFMD retrieval configuration details and it is therefore an inappropriate self-reference in the context of that paragraph ("verifying or improving operational products...", sensitivity to "details of the algorithm setup").*

**Authors:** Section 3.2 actually provides significant new information. In Hachmeister et al. 2022 issues of GMTED2010 over Greenland were identified using regional ICESat-2 data. The

update here is different in several aspects: it uses a different data set, namely the Copenicus GLO-90 DEM, and it is applied globally. In particular, the discussion of the spatially resolved impact by means of global maps is important and new. The example of Greenland (Figure 4) is shown because it is the region of the largest differences on the globe when updating the DEM. It has to be checked that the Greenland issue is also resolved when using GLO-90 (instead of ICESat-2 as in Hachmeister et al.). In fact, the methane distribution looks even more homogeneous with GLO-90 compared to ICESat-2. The uniqueness of this section is highlighted more clearly in the revised version.

Buchwitz et al. 2017 was intended to serve as a reference for the general desirability of an ensemble approach in terms of assessing the robustness of specific results regardless of the exact instrument or data set. Since this seems to be confusing and misunderstandable in this context, the citation will be removed.

**Reviewer:** *3) The paper reaches substantial conclusions with respect to the performance enhancement due to the new destriping method and adapted cloud-filtering, but the work on the polynomial fit parameter update (section 3.1) does not go beyond a qualitative discussion of a case-study and that section requires more attention (see specific comments).*

**Authors:** Section 3.1 is completely revised also discussing the improvement in fit quality and the global impact of the increase of polynomial degree quantitatively and its relation to the calibration for methane. The impact is also compared to the operational product where this update seems to be more significant. This is highlighted by the example of the Taymyr region in Northern Siberia. For TROPOMI/WFMD the impact is typically small and is largest for the Etosha pan. Further notable differences in Northern Africa and on the Arabian peninsula can be largely resolved by the subsequent calibration instead when using a quadratic polynomial. After calibration, the Etosha pan is by far the region of most significant performance improvement. This is the reason why it was the region of choice that was explicitly analysed in the original version. In that sense, it was far more than just an arbitrary case study. All these aspects are made more clear and discussed in more detail in the revised version (see also answers to specific comments).

**Reviewer:** *4) The article does not compare TROPOMI/WFMD results to the operational TROPOMI XCO and XCH4 products. I think that such a comparison would be a necessary addition to this manuscript.*

**Authors:** Although we do not think that this is a necessary addition, as it was already done in the previous algorithm paper, we added comparisons and discussions with the operational product in appropriate sections of the manuscript, e.g. in terms of impact of polynomial degree increase, magnitude of calibration correction, or striping artefacts.

**Reviewer:** *5) Please comment if TROPOMI/WFMD and its post-processor will be made available to fellow scientists, which would greatly profit traceability of this work.*

**Authors:** It is not planned to make the complete source code available, but we acknowledge key python packages relevant in the implementation of the random forest and destriping code in the revised version.

**Reviewer:** *6) The authors occasionally go into detail on the performance of XCO with v1.8*

*of their algorithm, but they do not systematically conduct their analyses for both CH4 and CO. I think the article would be much clearer if CO was either dropped entirely from this work or if equal emphasis was placed on the two molecules.*

**Authors:** The focus is on $XCH_4$ as the improvements are more relevant in terms of meeting the mission requirements. As XCO is more variable in the atmosphere the requirements are considerably relaxed. Most updates affect XCO in the same way (with the exception of the machine learning calibration). Since XCO is simultaneously retrieved with $XCH_4$, it is not dropped entirely but discussed in more detail in the revised version, where appropriate.

**Reviewer:** *7) The abstract should be edited to mention the TCCON analysis and the considerable improvement in filtering clouds above water surfaces.*

**Authors:** This is included in the revised version.

**Specific comments**

**Reviewer:** *Line 12: "machine learning calibration... has been optimised."; add a short statement describing the implemented updates.*

**Authors:** Has been included.

**Reviewer:** *Line 26/28: This sentence may be misunderstood in a way that the lifetime of methane is 9 years shorter than the lifetime of CO2. Please rephrase.*

**Authors:** Has been changed to "Since $CH_4$ (with a lifetime of about 9 years) is considerably shorter-lived in the atmosphere than $CO_2$, ...".

**Reviewer:** *Line 33: This occurs both through natural processes and resulting from human activities. -> This occurs both through natural processes and human activities.*

**Authors:** Has been changed.

**Reviewer:** *Line 49: ...TANSO FTS aboard GOSAT ... retrieves CO2 and CH4, ... -> ... measures/observes CO2 and CH4 absorption lines, ...*

**Authors:** Has been changed.

**Reviewer:** *Line 63: add citations to ATBDs for both operational products:*
*https://sentinels.copernicus.eu/documents/247904/2476257/Sentinel-5P-TROPOMI-ATBD-Carbon-Monoxide-Total-Column-Retrieval.pdf*
*https://sentinels.copernicus.eu/documents/247904/2476257/Sentinel-5P-TROPOMI-ATBD-Methane-retrieval.pdf*

**Authors:** Has been added.

**Reviewer:** *Line 64: add citation to https://amt.copernicus.org/preprints/amt-2022-255/*

**Authors:** Has been added.

**Reviewer:** *Line 100-105: To make this sentence more readable I suggest breaking it into two sentences.*

**Authors:** Has been done.

**Reviewer:** *Line 114: Is the only update a change with respect to the gridding of the underlying data? Please explain in the text.*

**Authors:** Yes, this is the only change and it is clarified in the text.

**Reviewer:** *Line 115: Was there a noticeable impact of the update in the meteorological reanalysis data on XCH4 results? Please elaborate.*

**Authors:** The theoretical improvements due to the better temporal and spatial resolution of the meteorological data are actually difficult to demonstrate and typically small, because noticeable impact is only expected when conditions change significantly on a small scale and/or in the short term. This note is added in the revised version.

**Reviewer:** *Line 115: v1.5 -> v1.2 (or if v1.5 is correct, what is v1.5?)*

**Authors:** v1.5 was the previous officially released data set before v1.8. For a better insight we have added a table summarising all versions with the respective differences in the algorithm setup.

**Reviewer:** *Section 3.1 makes the case that increasing the degree of the polynomial which approximates the spectral baseline (from 2 to 3) is a general improvement to the XCH4 v1.8 product. I find that this section would be significantly more robust and less of a qualitative discussion if the following updates were made.*

*a) The motivation of this section is to highlight changes in the spectral fitting procedure, yet no actual spectral analysis is presented here. The proposed fit updates would be much more convincing if the effecte on fit residuals was displayed, and if fit statistics (Chi2/RMS, convergence quality, etc.) for retrievals with and without the polynomial upgrade (in the Etosha pan and elsewhere) were included in the manuscript.*

**Authors:** We have included a spectral analysis discussing fits and residuals with quadratic/cubic polynomial in- and outside of the pan, as well as maps of the root mean square of the fit residuals demonstrating that the fit quality over the pan improves significantly with a cubic polynomial and becomes comparable to the surroundings of the pan, while there is no significant change outside of the pan.

**Reviewer:** *b) According to https://earthobservatory.nasa.gov/images/147221/cycles-of-wet-and-dry-in-etosha-pan, the wet season in the Etosha Pan occurs between October and March. The authors argue that wetland-related CH4 enhancements are visible in the images of the second row in Figure 2. However, this appears to be a measurement taken during the dry-season. Additionally the true color images from VIIRS do not show any obvious inundation in the region as far as I can tell. I suspect that the residual XCH4 enhancements in Figure 2c and 2f are therefore still retrieval artefacts. Spectral signal levels can probably be used to detect inundated TROPOMI pixels. Please explain what your arguments are based on and why you think the enhancements detected here are real.*

**Authors:** It is true that both days shown in Figure 2 are in the dry season, but from two different years (2019 and 2020), which differ significantly in terms of flooding. Due to abundant rainfall in the second half of the wet season (December 2019 - March 2020) the eastern part of the pan stayed inundated until end of July in the year 2020, well into the dry season. In contrast, the pan was entirely drained during the dry season 2019. Consistent with this, there is a methane enhancement over the eastern pan in 2020 that is not observed in 2019 in the TROPOMI/WFMD v1.8 product, while you see the extent of the complete pan on both days in the v1.5 XCH$_4$ (based on a quadratic polynomial in the fit).

In summary, the interpretation that the distribution over the pan is more realistic when using a cubic polynomial (apart from the better fit quality) is based on the fact that the shape of the retrieved XCH$_4$ enhancement for single overpasses is more variable. While the enhancement reflects the extent of the pan virtually always in the case of a quadric polynomial, it changes with meteorological conditions in the cubic case. For instance, the link between inundation and methane enhancement appears more evident in the latest product version.

As shallow inundation is hard to see in the VIIRS true color images the images are replaced by a VIIRS false colour band combination (Red = M3, Green = I3, Blue = M11) distinguishing different water states and thus enabling flood mapping in the revised version where the partial flooding of the day in 2020 is more obvious. The flooding can be traced in time here: `https://worldview.earthdata.nasa.gov/?v=13,-21,20,-17&l=VIIRS_SNPP_ CorrectedReflectance_BandsM3-I3-M11&lg=true&t=2020-05-16-T07%3A49%3A51Z`

*Reviewer: Line 132: How do you know what a realistic level of XCH4 enhancement is over the Etosha pan? Has this been studied before? If not, please reword.*

**Authors:** The text is reworded in a more qualitative sense (along the lines of the previous answer) to make clearer what we mean by "more realistic": better fit quality similar to the surroundings and more variability of the XCH$_4$ enhancement instead of just reflecting the extent of the pan for every single overpass as before.

*Reviewer: c) What is the significance of this spectral fit update in view of the post-processing steps following the retrieval, especially the machine learning calibration? How does the machine learning calibration affect the Etosha pan enhancements? Are plots in 2c, 2f prior to or after application of the machine learning calibration? What would 2c, 2f look like without the update in the polynomial, but with the updates in the machine learning calibration?*

**Authors:** The plots in Figure 2 are after application of the machine learning calibration. We included an additional figure demonstrating the impact of the polynomial update globally before and after calibration including a zoom on the Etosha pan, which is the region of largest impact. Typically the impact is small and most of the significant changes can alternatively be achieved by the calibration. In contrast, however, the differences/improvements over the Etosha pan cannot be entirely achieved through calibration justifying the implementation of the increased polynomial degree. This finding is also confirmed by the extended Figure 1 also showing v1.2 for comparison. The differences between v1.2 and v1.5 are due to the improved calibration (only resolving part of the enhancement over the pan) and the differences between v1.5 and v1.8 are mainly due to the increase of polynomial degree. These discussions are included in the revised version.

**Reviewer:** *Section 3.1 (Polynomial fit parameters) contains work that is conceptually similar to https://amt.copernicus.org/preprints/amt-2022-255/. Please include that paper as a reference and add a few remarks concerning that article's conclusions in view of your results.*

**Authors:** When our manuscript was written and submitted the mentioned preprint was not available yet. Of course, we have included it now as a reference and discuss the different impacts of the change on the products. The changes due to the adjustment of the polynomial degree seem to be less significant for TROPOMI/WFMD than for RemoTeC due to the alternative algorithm setup, e.g. with differing fitting windows and bias correction schemes. An example of this is the artefact in the Northern Siberian Taymyr region that occurs in the operational product, which was also misinterpreted as genuine large methane emissions from carbonate rock formations. While amt-2022-255 demonstrates that the original strong enhancement in the operational and RemoTeC scientific product is not reproduced when using a cubic polynomial, a corresponding distinct enhancement in this magnitude is not observed in the TROPOMI/WFMD products, even in the previous setup with a quadratic polynomial. We included a corresponding figure and discussion in the revised version.

**Reviewer:** *Line 119-120: "… it has been noted …" - by whom?*

**Authors:** The wording is changed in the revised version.

**Reviewer:** *Figure 2: What is algorithm version v1.5? It has not been introduced earlier. How does it differ from v1.2 (which I believe corresponds to your work published in Schneising et al. AMT 2019)? For clarity, I think it would be good to use v1.2 (Schneising et al 2019) for 2b and 2e.*

**Authors:** v1.5 was the previous officially released data set before v1.8. For a better insight we have added a table summarising all versions with the respective differences in the algorithm setup in the revised version. In the context of the discussion of the polynomial degree, v1.5 is more appropriate here, because the changes over the pan between v1.5 and v1.8 are mainly due to the increase of polynomial degree, while the changes relative to v1.2 also include significant improvements of the calibration. We have added v1.2 to Figure 1 and a corresponding discussion instead.

**Reviewer:** *The reflectance spectra from Moreira et al. 2014 and Tayebi et al. 2017 are a good find. I think it would be very informative to show a plot of WFMD residuals in comparison to those spectra.*

*I also suggest fixing the spectral baseline polynomial to a shape similar to the spectra from Tayebi et al. and Moreira et al. and observing what effect that may have on the retrievals.*

**Authors:** These spectra are only examples of possible interferences, which are roughly in this spectral range, and are not representative for the very specific soil types of the Etosha pan (calci sodic Solonchaks to sali calcic Solonetzs derived from Andoni sandstone or siltstone). Since the spectral albedo is generally not known, one cannot use a spectral database of soil types in the fit but has to rely on the polynomial. The spectral analysis added in the revised version suggests that a cubic polynomial approximates the (unknown) spectral albedo of this particular soil type sufficiently well (but a quadratic does not).

**Reviewer:** *Do surface reflectance induced biases, as observed for XCH4 retrievals, exist for*

*XCO in your algorithm?*

**Authors:** There is no identified obvious surface reflectance induced biases in the XCO product (see maps of yearly averages in the manuscript). A similar calibration of XCO is not necessary to achieve the mission requirements since a potential albedo-induced bias of the same percentage magnitude as for $XCH_4$ would not be significant due to the considerably higher variability of XCO. This is mentioned in section 2 of the revised version. As a consequence, the XCO requirements are an order of magnitude relaxed compared to $XCH_4$. The validation confirms that the mission requirements are actually achieved without calibration of XCO. Nevertheless, the potential for further improvement of the XCO product will be further investigated in the future.

**Reviewer:** *Line 184-186: Which dataset did you use for the surface roughness feature? Please add a reference in the manuscript.*

**Authors:** Surface roughness is determined from the respective DEM by using the standard deviation of the high resolution data within the considered gridboxes. This is clarified in the manuscript.

**Reviewer:** *With regard to the post-random-forest 3-step quality filter: What is the rationale of filtering for these three metrics separately instead of including RMS and spectral shift/squeeze as features in the random forest? Please clarify.*

**Authors:** The post-random-forest filter is an heuristic approach addressing outliers caused by residual issues not explicitly considered (yet) in the "truth" used in the training of the random forest classifier, such as specific scenes with intense aerosol exposure or other exceptional scenes with reduced fit quality relative to scenes with similar radiance for unknown reasons. RMS is already a feature in the random forest classifier but the basis of the training of the quality filter is VIIRS cloud information. Therefore, not all causes of relative poor fit quality can be learned and the residual outliers in terms of fit quality are probably not due to cloud contamination. Spectral shift/squeeze is rather considered relatively on a daily basis than through absolute thresholds in the random forest classifier to ensure that a potential future spectral drift due to degradation cannot cause unnecessary data loss. To integrate other issues in the random forest in the future one would need a reliable "truth" representing the respective issue. Since this post-random-forest filter is only about the remaining outliers it is impractical to identify and explicitly consider all conceivable issues in the training of the random forest classifier and the presented heuristic approach seems to be a satisfactory solution. This is better explained in the revised version.

**Reviewer:** *Generally, does the random forest still consider the 25 features listed in Schneising et al., AMT, 2019? Have you updated any other configurations of the machine learning code? Please explain in the text, and give some more introduction to your random forest set-up and updates you have made (e.g. near line 180).*

**Authors:** The random forest setup of the different versions is summarised in Table 1 of the revised version. For example, there are 26 features used in the random forest classifier since v1.5: the 25 features listed in Schneising et al., 2019 + surface roughness. The other update is the training data set: 30 (random) days instead of 16 and more ocean scenes in v1.8. At the beginning of Section 3.3.1, the updates are described in more detail and reference is made

to this table in the revised version (see also answer about the training data set below).

**Reviewer:** *Which features are most important for classifying clouds over water in your random forest classifier?*

**Authors:** The random forest is constructed for all traning data together not explicitly distinguishing between land and water. As a consequence, there is also no distinction in the statistical analysis of the most important features by land and water. The top 5 overall features are as in Schneising et al., 2019 with slightly changed order of importance and coefficients of determination.

**Reviewer:** *Line 184: You almost doubled the training data-set: Are the new training data exclusively ocean scenes? How did you select the new data for training?*

**Authors:** This is made clearer in the revised version: In v1.5 the training data is extended to 30 randomly chosen days from end of April 2018 until end of 2019 since v1.5 (compared to the original 16 days in v1.2). A total of 5 million measurements are selected for each day including all land data, all inland water data, and all ocean data passing the quality filter; the remaining amount is randomly sampled with bad quality ocean scenes. In v1.8, the quality filter is further improved using 18 million additional (bad quality) ocean scenes equally distributed over the 30 randomly chosen days when training the random forest classifier.

**Reviewer:** *Please go into detail why the filter underperforms over water. Intuitively, shouldn't these scenes be advantageous for the cloud filter, because the underlying surface is more homogeneous? A Figure showing the performance of the filter (and its challenges) over water would be very helpful here.*

**Authors:** As described in the previous answer (and the revised version), the main focus of the quality filter has been on land scenes. The selection of the training dataset in v1.8 corresponds better to the actual partition between land and water scenes. Furthermore, the underperformance does not affect water scenes in general, but mainly scenes in the Arctic ocean in summer. Due to the low absolute number of good measurements in the Arctic ocean (no sun glint), a high percentage error rate can be obtained by relatively few misclassifications. We have added a Figure demonstrating the performance of the quality filter compared to VIIRS over the Artic for an example day in July to the revised version.

**Reviewer:** *Paragraph from lines 216-230: How great is the loss of "good" measurements? If the numbers in Figure 5 are upper limits, how far from the upper limits is the actual loss (for the different cases you study in Fig. 5)?*

**Authors:** The numbers have been revised accordingly.

**Reviewer:** *Do you have any plans to include a cloud-shadow filter in your algorithm?*

**Authors:** This is not explicitly planned so far, but it is worth considering.

**Reviewer:** *What impact do shadowy scenes have on the ensemble of your retrievals?*

**Authors:** This has not been studied in detail. To a first approximation, it should behave similarly to a lower albedo. Therefore, potential shadow issues may also be resolved by the

quality filter and/or the calibration.

**Reviewer:** *Line 216: "increases" -> increase*

**Authors:** We think that "The ... tightening ... increases the precision ..." is correct.

**Reviewer:** *Line 233-234: I thought the introduction of the third-order baseline fit (section 3.1) was meant to resolve the albedo bias issue? If it remains, why bother making that adjustment if you are re-calibrating XCH4 in the post-processor anyway?*

**Authors:** It is the other way round: The third-order baseline fit is meant to resolve the remaining issues that cannot be resolved by the calibration. That becomes clearer through the significantly improved and extended Section 3.1. Overall, one has to distinguish here between two fundamentally different issues: 1) Biases due to low signals (likely an instrumental issue as the quite different RemoTeC algorithm shows very similar behaviour) addressed by the calibration and 2) Issues due to the "shape" of spectral albedo within the fitting window (independent of the magnitude of the signal) addressed by the third-order baseline (or altenatively also by calibration in most cases). As the calibration is not always sufficient to resolve all issues of type 2), the third-order baseline fit is needed in addition. This has also been clarified in Section 3.3

**Reviewer:** *Why are you not conducting a machine learning calibration for XCO? Are the retrievals better? If yes, how are they better? Can the XCH4 "calibration" inform corrections to XCO?*

**Authors:** A similar calibration of XCO is not necessary to achieve the mission requirements since a potential albedo-induced bias of the same percentage magnitude as for $XCH_4$ would not be significant due to the considerably higher variability of XCO (see also above discussion). As a consequence, the XCO requirements are an order of magnitude relaxed compared to $XCH_4$. The validation confirms that the mission requirements are actually achieved without calibration of XCO. Nevertheless, the potential for further improvement of the XCO product will be further investigated in the future and it may be possible to determine a XCO correction based on the $XCH_4$ calibration in terms of albedo-induced biases.

**Reviewer:** *I realise that it is standard practice in trace gas retrievals to carry out bias corrections. However, I have not seen a post-processing correction, or "calibration", that brings in information from an a priori/model dataset again. This seems to defy the purpose of running a retrieval. The way this section is currently written makes it very hard for the reader to understand what impact the calibration has on the WFMD results and it raises many questions.*

*For example, how does the calibration affect XCH4 values in retrieved methane plumes?*

*How does it impact CH4 inversions on regional scales?*

*How far to the prior XCH4 does this correction pull the retrieved values (how does the distribution of retrieved values change)?*

*How does the correction vary by latitude and season? Please elaborate in the manuscript.*

**Authors:** The purpose of the calibration is not to bring in information from a model and it does not pull the retrieved values to a prior. The model is only used as a coarse approximation

of the background to determine statistical discrepancies depending on albedo (see Section 2 and revised Section 3.3). After learning of the respective statistical relationships on a temporally and spatially limited dataset in training, the climatology is not used anymore and the calibration is then performed only using the listed intrinsic parameters of the retrieval, in particular the retrieved apparent albedo. This process is called "calibration" because the corrected biases due to low signals are likely an instrumental issue (e.g. due to a radiance-dependent zero-level offset shift of the detector). The calibration does not defy the purpose of a retrieval, it is part of the retrieval. This is explained again in more detail in Section 3.3 of the revised version. In particular, there is no attenuation of methane plumes or gradients by mixing in a smooth prior what may be suspected here by the reviewer. On the contrary, the calibration has no effect at all on methane plumes since they typically do not cause any albedo change. Regional inversions are improved by the calibration because potential biases due to a varying background albedo are resolved. We have added seasonal global maps demonstrating the magnitude (which is broadly similar to the correction applied in the RemoTec/operational algorithm) of the calibration and mainly reflecting albedo features. Since the calibration is independent of TCCON data, the conclusive power of the subsequent validation, which does not indicate any problems, is not limited.

**Reviewer:** *Line 240: Are you retrieving XCH4 from spectra at SZA>70? Are you taking the non-planar nature of the atmosphere into account for those cases? Do you trust your retrievals at such SZA values and what do they look like?*

**Authors:** Yes, we use SCIATRAN in pseudo-spherical atmosphere mode to (pre-)calculate the forward model, i.e., the ray tracing for the direct solar beam is performed assuming a spherical atmosphere. This information has been added to the manuscript. This setting provides sufficient accuracy of the modelled radiation for solar zenith angles up to about $92°$. Furthermore, the SZA is a feature in the quality filter and the calibration. In summary, we generally trust our quality filtered and calibrated retrievals with SZA$< 75°$ and there are no obvious issues with high solar zenith angles when validating the products with TCCON (e.g. at Arctic TCCON sites). Nonetheless, these are of course challenging conditions and related issues cannot be completely ruled out under specific conditions.

**Reviewer:** *Line 253: Please explain why you subtract specifically 5 ppb (given that the SLIMCH4 bias at the three northernmost TCCON sites is greater than 5 ppb)? What is the mean Arctic bias and what do you consider "typical" bias levels?*

**Authors:** The wording has been changed. The biases at the Arctic sites are consistently positive ranging from about $10\,\mathrm{ppb}$ (East Trout Lake) to $20\,\mathrm{ppb}$ (Ny Alesund). Typical biases at other sites are about $\pm5\,\mathrm{ppb}$ and occasionally reach $10\,\mathrm{ppb}$. Therefore, a compromise of $-5\,\mathrm{ppb}$ for the Arctic region was chosen to account for the systematic positive bias, while avoiding potential overcompensation.

**Reviewer:** *Line 237: is $r_{cld}$ the cloud flag from your cloud filter? Or is it from VIIRS?*

**Authors:** This is the ratio of measured to reference radiance for selected strong $H_2O$ absorption lines as described in Schneising et al., 2019. This has been clarified in the revised version. VIIRS information is only used in the training and afterwards the quality prediction is independent of VIIRS.

**Reviewer:** *When you include the across-track index in the calibration, what impact does that have on the striping pattern in XCH4? What is the reason to include the across-track dimension index if it does not sufficiently destripe the images?*

*Do you see any possibility to get the destriping done within this calibration scheme so that you do not need to run the wavelet procedure?*

**Authors:** The rationale of the inclusion in the calibration is to take into account recurring systematics, e.g. potential smooth biases induced by viewing zenith angle or temporally constant striping patterns. Actually, this improves the striping to some extent. The dedicated wavelet approach is applied orbitwise and therefore also takes temporally variable striping patterns into account. Thus, we do not think that destriping can be entirely achieved by calibration, especially in a shallow implementation. This discussion is added to the introduction of Section 3.4.

**Reviewer:** *Figure 6: How did you chose the regions used in the training of the machine learning regressor? It looks like there is a latitude band missing (tropics) - why did you exclude that? At which point in the calibration procedure do TCCON measurements actually come into play (other than being a validation source for the climatology you are using)? If the correction does not explicitly use TCCON data, please remove the TCCON stations from the map, because it misleads the reader to think TCCON data go directly into the correction. Please explain in the manuscript.*

**Authors:** The regions are selected to cover the whole range of albedo values and all possible viewing geometries, as these are important features used in the calibration. This is made more clear in the revised version. Furthermore, the regions are chosen close to TCCON sites, which are used in the validation of the climatology. As a consequence, the quality of the climatology within the regions is assumed to be good and that was also the reason to include the TCCON sites in the figure. As this seems to be misleading, we removed the stations from the map in the revised version as TCCON measurements only come into play as a validation source for the climatology, the correction is otherwise independent of TCCON. The South Atlantic region contains part of the tropics. There are no TCCON sites closer to the equator and the data density of quality filtered data near the equator is low due to frequent cloud coverage and low albedo.

**Reviewer:** *Figure 8: "Coiflet wavelets" are introduced in the caption for the first time. Please also add a short explanation in the main text before referring to coiflets in this plot.*

**Authors:** We added a short explanation and reference in the main text and a figure to the appendix.

**Reviewer:** *Lines 351-353: An analysis of the temporal development of atmospheric CO concentrations in comparison to other measurements would be very valuable here as well. Perhaps https://doi.org/10.1016/j.rse.2020.112275 or https://www.epa.gov/air-trends/carbon-monoxide-trends could be starting points for a comparison.*

**Authors:** A comparison of the temporal development of CO with other data sets would be valuable but is less straightforward than for methane, because CO has a significantly shorter lifetime, is less mixed and more variable. This makes it much more difficult to find data sets

that are indeed suitable for comparison, e.g. in terms of spatial sampling, vertical sensitivity or covered time period. Different regions may have significant different CO growth rates. In the case of CO, a regional break-down would be more appropriate, but that would go beyond the scope of this (already very comprehensive) paper. That would easily fill a paper of its own. For example, Buchholz et al., 2021 does not fit from the period and the EPA trends refer to the United States only.

***Reviewer:*** *Line 455: Change citation Dlugokencky: Lan, X., K.W. Thoning, and E.J. Dlugokencky (2022): Trends in globally-averaged CH4, N2O, and SF6 determined from NOAA Global Monitoring Laboratory measurements. Version 2022-10, https://doi.org/10.15138/P8XG-AA10*

*This is the recommended citation (see bottom of page* `https: // gml. noaa. gov/ ccgg/ trends_ ch4/ `*)*

**Authors:** Has been changed in the revised version.

---

## Author Comment (AC2)

**Final response to referee comments on paper amt-2022-258**

First of all, we would like to thank reviewer #1 for his/her constructive comments, which helped to improve the manuscript. Below we give answers and clarifications to all comments made by the referee (repeated in italics).

**Anonymous Referee #1**

**Specific comments**

**Reviewer:** *The paper made extensive comparisons between v1.5 and v1.8 of the TROPOMI/WFMD algorithm. The previous algorithm paper (Schneising et al., 2019) is focused on v1.2, and it appears that a detailed description of v1.5 has not yet been published. It would be helpful to add a table that summarizes the differences between the three versions (v1.2, v1.5, and v1.8).*

**Authors:** A corresponding table is added to the revised version summarising the respective differences in the algorithm setup of the different versions.

**Reviewer:** *Figure 1 and section 3.1: the increase of polynomials in fitting is an updated applied to all retrievals. Can the authors include some results for other areas?*

**Authors:** Section 3.1 is completely revised also discussing the Taymyr region in Northern Siberia and the global impact of the increase of polynomial degree quantitatively using a global map of the induced differences. For TROPOMI/WFMD the impact is typically small and is largest for the Etosha pan. That was the reason to focus on this region.

**Reviewer:** *Figure 3a: there appears to be a gradient along 60 N, especially over Siberia. Is there any explanation for this?*

**Authors:** This gradient can be attributed to the GMTED2010 data. This is a known limitation of GMTED2010, as the DEM is composed of various datasets and the main dataset used in GMTED2010, the Digital Terrain Elevation Data v2 (DTED 2), is only available for latitudes between 60°N and 56°S. The topography information at higher latitudes is based on older data with lower spatial resolution. This discussion is added to the revised version. That the high latitude difference pattern is an issue of GMTED2010 and not of GLO-90 can also be seen in a new Figure showing the methane distribution for several TROPOMI/WFMD versions over Northern Siberia demonstrating that v1.8, which is based on GLO-90, exhibits the most homogeneous methane distribution in this region.

**Reviewer:** *Figure 4: perhaps it would be useful to compare the same version of the algorithm with the DEM model as the only difference.*

**Authors:** v1.5 was the previous officially released data set before v1.8. Thus, we think that the comparison of these officially released products is most expedient. Furthermore, the other changes are negligible compared to the impact of the DEM change over Greenland, which can occasionally exceed 100 ppb. The difference pattern is reflected 1:1 in the inhomogenities of

v1.5 at coastal Greenland as can be seen in Figure 3b (Figure 6b of the revised version). For example, there are no significant changes due to the increase of the polynomial degree over Greenland as can be seen in the newly added figure demonstrating the global impact of this modification (see also above).

***Reviewer:*** *Section 3.3.1/3.3.2, does the change in TROPOMI spatial resolution have any effects on the trained models for cloud screening and bias correction? Do the models need to be re-trained to account for the change in resolution (and also the cross-track index)?*

**Authors:** There is only one common quality filter and calibration for both periods before and after the change of spatial resolution and thus no re-training. The relevant time periods for training the quality filter and calibration (subsets of 2018 and 2019 covering both footprint sizes) are added in the revised manuscript. The footprint size change in August 2019 only (slightly) changes the along-track dimension of the measurements, the across-track expanse does not change. Therefore, there is no issue with the across-track index used in the calibration associated with this change.

***Reviewer:*** *Lines 191-193: how were the parameters a, b, and c determined?*

**Authors:** These parameters were determined empirically to distinguish between typical values of the root mean square of the fit residual (as function of the sun-normalised radiance) and outliers for whatever reason, such as specific scenes with intense aerosol exposure exhibiting reduced fit quality relative to scenes with similar radiance. This is better explained in the revised version.

***Reviewer:*** *Section 3.3.2, given that the cross-track index is an input to the trained model, would one expect the striping to be eliminated or reduced by the bias correction procedure?*

**Authors:** The inclusion of the across-track index in the calibration accounts for recurring systematics, e.g. potential viewing zenith angle issues or temporally constant striping patterns. As a result, striping is improved to some extent but complete destriping, in particular with respect to temporally variable striping patterns, cannot be entirely achieved by a shallow calibration. Therefore, the remaining vertical stripes in the satellite data after calibration are efficiently removed orbitwise in v1.8 by combined wavelet-Fourier filtering. This discussion is added to the introduction of Section 3.4.

***Reviewer:*** *Section 3.4 – is the destriping scheme run after the bias correction? Please clarify.*

**Authors:** It is run after calibration. This is clarified in the revised version (see also answer to previous comment).

***Reviewer:*** *Line 296: it would be helpful to briefly introduce Coif16 to readers who are less familiar with the method.*

**Authors:** We added a short explanation and reference in the main text and a figure to the appendix.

***Reviewer:*** *Table 1: Why are the random error and especially systematic error for v1.2 different from the values given in Schneising et al. (2019)? Also it appears that the systematic error in this paper follows a different definition?*

**Authors:** There are 5 additional TCCON sites involved in the validation and the validation time period is extended by 1 year. Therefore, exactly the same numbers cannot be expected. The definition of systematic error has slightly changed also taking seasonal biases into account. These differences are clarified in the revised version.

---

## Author Response (AR2)

Bremen, January 16, 2023

**Letter to the Editor of paper amt-2022-258**

Dear Joanna,

on behalf of all co-authors I have prepared this document, which provides the point-by-point responses to the suggestions of reviewer #2. The corresponding changes (and very few other minor stylistic revisions of the text) made in the manuscript are highlighted in the attached track-changes file.

Best regards,

Oliver

**Response to referee comments on revised paper amt-2022-258**

Below we give answers and clarifications to all comments made by the reviewer #2 (repeated in italics).

**Minor comments**

***Reviewer:** 1) In light of your recent publication on using ICESat-2 for TROPOMI retrievals, it would be interesting for readers to know if there is any significant difference between GLO-90 and ICESat-2 over Greenland. Please add a statement that puts the current algorithm update (GLO-90) into context with the retrievals you obtained with ICESat-2: Is GLO-90 the recommended DEM for retrieving methane over Greenland? This section may naively come across as if Hachmeister et al. was now outdated with respect to the DEM data source; please clarify.*

**Authors:** There is a similar improvement over Greenland when using GLO-90 or ICESat-2 instead of GMTED2010. If it is (only) about Greenland, you can use both DEMs. The added value of GLO-90 is its global consistency, which additionally permits to resolve further potential DEM inaccuracies and related retrieval biases elsewhere. This has been clarified.

***Reviewer:** 2) Is the spectral range of the TROPOMI WFMD fitting windows specified anywhere in the text of the manuscript (has it changed since v1.2?)? Please clarify. If the fitting windows are identical to the spectral range(s) shown in Fig. 5 [a,b], please discuss in the manuscript why you do not use the whole CH4 absorption band (as measured by TROPOMI) in your fits. The fact that these spectral windows are relatively narrow may explain why WFMD is less susceptible to some of the observed albedo bias features in comparison to the operational TROPOMI retrievals, which cover a wider spectral range. Lower order polynomials can approximate albedo-induced structures in the spectral baseline well, if the spectral range is sufficiently small and the spectral signature of the surface reflectance is of a broadband nature (i.e. variations on the scale of 10s of nanometres). Please consider adding a statement about this in the manuscript, for example in line 146-147. The way this sentence is written, appears to suggest that there is something fundamental about either algorithm that leads to better or worse retrieval quality. Please rephrase to emphasise that it is the choice of spectral windows that has a significant impact on different aspects of the retrieval; among them spectral information content, the ability of spectral albedo features to affect retrieved methane concentrations, etc..*

**Authors:** The spectral fitting windows are unchanged since v1.2 and span the spectral range shown in Figure 5. They were optimised based on an error analysis (also including typical spectral albedo scenarios) of simulated measurements. It has been found that systematic errors are minimised if we do not use the whole $CH_4$ absorption band as measured by TROPOMI. This finding and the connection between choice of fitting windows and spectral albedo biases is discussed in a new paragraph starting in line 146:

"Overall, the changes due to the adjustment of the polynomial degree seem to be less significant for TROPOMI/WFMD than for RemoTeC. The lower susceptibility of TROPOMI/WFMD to some of the observed spectral albedo bias features is primarily attributed to the narrower

spectral fitting windows in comparison to the RemoTeC retrievals, which cover a wider spectral range. If the spectral range is sufficiently small, it is easier to approximate albedo-induced structures in the spectral baseline by lower degree polynomials. To retrieve $CH_4$ and CO simultaneously as accurately as possible, the TROPOMI/WFMD spectral fitting windows were optimised based on an error analysis of simulated measurements (also including spectral albedo scenarios of typical surface types) resulting in the windows 2311–2315.5 nm and 2320–2338 nm (Schneising et al., 2019). For instance, it was identified that it is beneficial with regard to systematic errors to exclude the strong methane absorption lines between the two fitting windows, although the associated loss of spectral information may lead to a slightly increased random error."

**Reviewer:** *3) You could consider moving section 3.4.1 to the appendix to make the main body of the manuscript more compact.*

**Authors:** Although section 3.4.1 is of technical nature, we prefer to keep it in the main text because it facilitates the understanding of what follows and thus enables a linear flow of reading without scrolling back and forth. We would like to reserve the appendix for brief additional information that is not absolutely necessary for understanding the main text.

**Technical suggestions**

**Reviewer:** *1) Line 225: "blocky" -> "discontinuous"*

**Authors:** Has been changed.

**Reviewer:** *2) Caption Fig. 1 "Juli" -> "July"*

**Authors:** Has been corrected

**Reviewer:** *3) Fig. 2: What temporal range of TROPOMI data was used for this plot? Please add in caption.*

**Authors:** The temporal range (2018-2020) is added to the caption.

**Reviewer:** *4) Please mention in the caption of Figure 5 that XCO images show yearly averages (which year(s)?).*

**Authors:** All maps in Figure 5 (including XCO) show a single satellite overpass from the same day. This is clarified in the caption.

---

## Author Response (AR3)

Bremen, January 18, 2023

**Letter to the Editor of paper amt-2022-258**

Dear Joanna,

the track changes version can be found under the link "Author's tracked changes" of the file upload on 16 January. As there are only a few revisions, it may not have been immediately obvious that the changes are highlighted in this version.

Best regards,

Oliver